# Impact of Membrane Modification and Surface Immobilization Techniques on the Hemocompatibility of Hemodialysis Membranes: A Critical Review

**DOI:** 10.3390/membranes12111063

**Published:** 2022-10-28

**Authors:** Simin Nazari, Amira Abdelrasoul

**Affiliations:** 1Division of Biomedical Engineering, College of Engineering, University of Saskatchewan, 57 Campus Drive, Saskatoon, SK S7N 5A9, Canada; 2Department of Chemical and Biological Engineering, College of Engineering, University of Saskatchewan, 57 Campus Drive, Saskatoon, SK S7N 5A9, Canada

**Keywords:** hemodialysis membrane, hemocompatibility, surface modification, performance improvement, membrane fouling, protein adsorption

## Abstract

Despite significant research efforts, hemodialysis patients have poor survival rates and low quality of life. Ultrafiltration (UF) membranes are the core of hemodialysis treatment, acting as a barrier for metabolic waste removal and supplying vital nutrients. So, developing a durable and suitable membrane that may be employed for therapeutic purposes is crucial. Surface modificationis a useful solution to boostmembrane characteristics like roughness, charge neutrality, wettability, hemocompatibility, and functionality, which are important in dialysis efficiency. The modification techniques can be classified as follows: (i) physical modification techniques (thermal treatment, polishing and grinding, blending, and coating), (ii) chemical modification (chemical methods, ozone treatment, ultraviolet-induced grafting, plasma treatment, high energy radiation, and enzymatic treatment); and (iii) combination methods (physicochemical). Despite the fact that each strategy has its own set of benefits and drawbacks, all of these methods yielded noteworthy outcomes, even if quantifying the enhanced performance is difficult. A hemodialysis membrane with outstanding hydrophilicity and hemocompatibility can be achieved by employing the right surface modification and immobilization technique. Modified membranes pave the way for more advancement in hemodialysis membrane hemocompatibility. Therefore, this critical review focused on the impact of the modification method used on the hemocompatibility of dialysis membranes while covering some possible modifications and basic research beyond clinical applications.

## 1. Introduction

Membrane surface characteristics have a significant impact on membrane hemocompatibility and antifouling properties because the blood is directly in contact with the membrane. It’s not surprising, then, that many research efforts are focused on membrane surface modification and it is the key technology in this field [1,2,3,4,5,6,7,8]. The goals of membrane surface modification are (1) reduction of undesirable membrane-human serum proteins interactions, which leads to blood activation, in addition to protein adsorption (membrane fouling) that degrades efficiency, and (2) enhancement of selectivity or even the generation of totally new separation capabilities [9,10,11,12,13]. In the process of membrane modification, various variables must be critically controlled, including, stability, homogeneity, roughness, process control, and acceptable cost, as well as the fine-tunable of functional groups, which is a significant problem [14]. The key challenges that can be addressed by surface modification for hemodialysis membranes are hemocompatibility and human serum proteins adsorption which result in the activation of immune system cascades and membrane fouling and its critical consequences for hemodialysis patients [15,16,17,18,19]. Membrane fouling is the deposit of blood proteins, platelets, and other blood constituents on a membrane surface or within its pores, resulting in a decrease in membrane performance and the efficiency of clearing toxins. The degree of membrane fouling caused by protein adsorption, denaturation, and aggregation is influenced by the interaction between membrane surfaces and blood constituents [6,11,15,20,21,22,23]. The hydrophilicity of membranes is a challenge in dialysis. Some researchers claimed that hydrophilicity has a crucial effect on fouling resistance [24,25,26,27,28,29,30,31,32], as the hydrophilic surface absorbs so much water that it prevents protein adsorption, and in some situations even completely prevents it, resulting in a more hemocompatible membrane.

Our research group has conducted extensive studies on membrane modification and its improved biomedical applications, particularly for hemodialysis membranes [19,33,34,35,36,37,38,39].

So, the current tendency is to create novel and more effective membrane materials to lower protein attachment improving hemocompatibility and toxins removal efficiency to increase dialysis performance. Alternative materials, on the other hand, are frequently found to be less stable and (unreasonably) expensive [40]. Surface modification is the process of changing the physical, chemical, or biological features of a membrane’s surface from those of the polymeric membrane. Polymer membranes with outstanding hydrophilicity and hemocompatibility can be made utilizing appropriate surface modification techniques [25,41,42,43,44,45,46,47,48]. Physical and chemical procedures are the two primary groups of these strategies. The classification of these significant groups is determined by whether or not a chemical bonding happens through the process. Generally, physical changes do not affect the chemical structure of the membrane. The use of grinding, polishing, and thermal treatment to change the surface characteristicsof a membrane, such as roughness, grain boundaries, andsize, is common in these procedures. Blending and coating, on the other hand, are the most prevalent physical-process-based approaches that affect the membrane chemical composition. In contrast to physical methods, chemical techniques such as grafting, ozone & plasma treatment, photochemical & radiation reaction, radical polymerization, click chemistry, and biomimetic treatment, definitely alter the final composition of the membrane surface [49]. This research builds on our early studies by presenting typical strategies for enhancing the antifouling property and hemocompatibility of the membranes, making them more successful in the hemodialysis procedure [19,35,37,43,50,51,52,53,54,55]. The most common methods are first described separately; then, in the next part, they are evaluated and their advantages and disadvantages are discussed.

## 2. Physical Modifications

Surface alteration can be accomplished through physical methods which do not require any chemical interactions. Modifications, such as polishing, grinding, thermal treatment, blending, and coating are easy and inexpensive ways to change the surface roughness, grain size, and grain boundary of membranes [56,57]. However, the application of physical modification is limited as they are better suited to flat membranes rather than hollow fiber membranes. Moreover, polishing and grinding processes are only applicable to inorganic membranes.

### 2.1. Polishing and Grinding

Polishing and grinding, which use abrasive materials like sandpaper and diamond powder to adjust the surface roughness of a membrane, were among the first methods of manipulating membrane surface physical attributes. R. Meghnani et al. [58] designed and developed novel ceramic membranes and polished them with a silicon carbide abrasive sheet for hemofiltration application. Clay ingredients were mixed and compacted to create the membrane. The membrane was made by combining clay components, compacting them, and thensintering them at temperatures ranging from 900 to 1100 degrees Celsius. To achieve a uniform smooth surface, the strong sintered membranes were carefully polished on both sides with a silicon carbide abrasive sheet. Based on the results of protein attachment tests, platelet adhesion, whole blood clotting time, blood coagulation time, and complement activation, this type of membrane demonstrated excellent hemocompatibility, clearly showing that polishing was a good technique for surface modification in inorganic membranes [4]. This procedure is easy and affordable, and no organic solvents or reagents are required; nonetheless, it is only suited for inorganic and flat membranes.

### 2.2. Thermal Treatment

Thermal treatment is a kind of physical technique, which involves heating the membranes over time. By heating a membrane, the physical structure will be altered, including the amount and size of pores, which will improve separation properties [59,60]. Not only is it inexpensive, easy to use, and eco-friendly, but it may also prove useful for the modification of hollow fiber membranes. J. Barzin et al. [61] used the dry-wet spinning method to synthesize hemodialysis (HD) hollow fiber membranes from polyethersulfone (PES) and poly(vinylpyrrolidone) (PVP) solutions in N,N-dimethylacetamide (PES/PVP = 18/3 and 18/6 by weight). Heat-treated hollow fibers (in the air at 150 °C for 5 min) exhibit significantly increased and decreased water flux in heat-treated water and air respectively. The AFM morphological study of the membrane internal and external surfaces revealed that the roughness decreased after heat-treatment, either in water or in air. Furthermore, SEM indicated that the membrane surface morphology changed after heat treatment. Increased surface smoothness (or decrease in roughness) resulted in increased UF membrane separation, showing that hollow fiber heated membranes are superior for hemodialysis because of their improved surface morphology. M. Gholami et al. [62] revealed similar results in another study, reporting that when the thermal treatment temperature rises to 230 °C, the pore size of PES hollow fiber membranes decreases. To explore the influence of heat treatment on ultrafiltration UF capability, PES hollow-fiber membranes were produced using the dry-wet spinning technique and then heated in an oven at various temperatures. Following research into the effects of heat treatment temperature and time, it was determined that a combination of 150 °C and 5 min generated the greatest results. Even though there was no obvious alteration in the hollow-fiber diameter, annealing was shown to affect the surface morphology of the membrane, as indicated by a drop in flux and an elevation in solute separation.

### 2.3. Blending

Blending is the physical mixing of two (or more) polymers to get desired characteristics. Blending modification is the most practical method for use in large-scale manufacturing. Membrane morphological properties are influenced by factors such as the casting solution (polymer and solvent type, and polymer to solvent ratio), producingconditions (temperature, method), and additives [63]. By blending various chemicals during membrane production, polymer membranes with hydrophilic surfaces and other beneficial characteristics can be manufactured without pre- or post-treatment. In addition, by using this technique, hollow fiber membranes may be modified, unlike most grafting methods, which modify flat sheet membrane surfaces. The most extensively utilized blending components to increase the hydrophilicity of the HD membrane are biocompatible hydrophilic polymers such as polyethylene glycol (PEG), oligo (ethylene glycol) (OEG), polyvinylpyrrolidone (PVP) and zwitterions polymers (ZWs) [63,64,65,66,67,68,69,70,71,72]. Sinha et al. [73] used the phase inversion approach to create a flat sheet asymmetric polymeric membrane using polysulfone (PSF) and N-methyl-2-pyrrolidone (NMP) as casting solvents (Figure 1). Polyethylene glycol methyl ether (PEGME) in three distinctmolecular weights was used as an additive polymerin the casting solution. The hydrophilicity of the modified membraneswasdemonstrated to be improved by raising the additive molecular weight (from 200 to 5000 Da) induced by the addition of additives to the membrane matrix as well as the creation of more pores. There was also a considerable rise in bovine serum albumin (BSA) rejection as well as flux.

Poly(lactic acid) (PLA) HD membranes with improved fouling resistance and hemocompatibility were developed by L. Zhu et al. [45] using poly(lactic acid)-block-poly(2-hydroxyethyl methacrylate) (PLA-PHEMA) copolymers as the blending additive. PLA-PHEMA block copolymer was made via reversible addition-fragmentation polymerization and was utilized as a hydrophilic additive to change PLA membranes using the non-solvent induced phase separation (NIPS) technology. PLA/PLAPHEMA membranes with high PLA-PHEMA content showed improved hydrophilicity, which resulted in boosted compatibility (reduced platelet adhesion, prolonged plasma recalcification, and decreased BSA adsorption) and antifouling qualities(better water flux recovery ratio (FRR) and improved fouling resistance). The findings revealed that PLA-PHEMA copolymers were excellent additivesfor improving the properties of PLA-based membranes used in hemodialysis.

O. Azhar et al. [74] used Polyvinyl Alcohol (PVA) and PEG as additives to improve the filtering performance and hemocompatibility of cellulose acetate (CA) HD membrane. With a higher concentration of the copolymer, the cross-sectional shape of the membranes altered significantly, revealing a spongy structure that improved moisture absorption. Increases in the concentration of PVA also increased the surface roughness. According to SEM images, the surface of the membrane exhibited asymmetrical pores and the mean size of the membrane pores has changed with changing the concentration, which impacts the membrane selectivity in removing toxins and rejecting proteins. These alterations revealed the critical involvement of PVA in the establishment of structure in the CA-PVA membrane. As a biocompatible hemodialysis membrane, CA-PVA looked impressive. The modified membrane is more biocompatible than the pure CA membrane, as evidenced by the 95% rejection of BSA, decreased platelet adsorption and hemolysis ratio, and low thrombus formation on the membrane surface. PVA appears to be an appealing material for constructing asymmetric porous membranes with good selectivity and permeability. In addition to having a high fouling resistance, PVA is extremely hydrophilic, has good pH stability, is highly mechanically durable, and shows excellent biocompatibility [75]. PVA can be employed as an additional polymer or a low molecular weight additive to modify membrane structure [76]. However, the presence of hydrophilic polymers such as PEG, PVP, and PVA on the dialysis membrane surface could pose serious problems in the HD process due to their slow elution from the membrane over time [77]. As a result, many strategies were used to increase the performance of the additive, such as usingdi and tri amphiphilic copolymers and nanocomposites with a variety of molecular configurations and structures, thermally cross-linking PVP, or tailored function of the polymers [78,79].

H. Song et al. [79] made a highly branched block copolymer poly (vinyl pyrrolidone)-block-poly (acrylate-graft-poly (methyl methacrylate))-block-poly-(vinyl pyrrolidone) (PVP-b-P(AE-g-PMMA)-b-PVP and utilized it in the modification of PES membranes. The PVP-b-P(AE-g-PMMA)-b-PVP chains were directed onto the membrane surface to enhance hydrophilicity and hemocompatibility, and AE-PMMA chains were intertwined with PES networks to prohibit copolymer leaching when immersed in water. The modified PES membranes had a better water contact angle (WCA), lower platelet adsorption, and a longer clotting time than the virgin membrane. In addition, the flux of PBS (or water) and the resistance to protein fouling were considerably enhanced. Sun et al. [80] reportedsilica-PVP nanocomposites as a blending additive for PES modification and discovered that the protein antifouling properties were superior to the PVP additive. PVP surface coverage and hydrophilic characteristics are higher in the PES membrane with a silica-PVP nanocomposite additive than in the PES membrane with just PVP leading to superior flow recovery and better fouling resistance.

Sulfonation is another chemical modification process that uses a sulfonation agent to add hydrophilic groups to polymer networks. Athira et al. [81] investigated blended PES membranes made of CA and sulfonated polyether sulfone (SPES) (Figure 2), and illustrated that SPES/PES and CA/PES blend membranes are more hydrophilic and hemocompatible than PES membranes. SEM analysis of the blend membranes revealed considerable alterations in membrane morphology and pores. Based on the protein fouling, platelet adsorption, and activated partial thromboplastin time (APTT) tests, SPES/PES membranes show better hemocompatibility than their equivalents.

Citric acid, which is commonly employed as an anticoagulant, is grafted onto the polymer surface as a synthetic additive. T. M. Liu et al. [82] used phase-inversion technology to develop a blend membrane with improved hemocompatibility and antibacterial properties utilizing polyurethane (PU) materials treated with citric acid and chitosan. A three-step reaction was effective in synthesizing the PES-PU-CA-CS membrane. Initially, a PU pre-polymer containing an isocyanate group was made using a one-pot approach, then grafted with citric acid andblended with PES to prepare a membrane for grafting chitosan via esterification and acylation processes. Surface and cross-section SEM images of PES, PES-PU-CA, and PES-PU-CA-CS membranes revealed no discernible variations, indicating that the blending PU-CA and grafting of CS did not affect the membrane structure. Based on the surface analyses, the PES-PU-CA membrane had a rougher surface compared to the PES membrane, possibly because of blending with PU-CA and that some outshoot, such as points and lines, developed on the PES-PU-CA-CS membrane, maybe due to grafted CS aggregation. For all membranes, a unique finger-like across-section morphology was developed, comprising aporous sub-layer and a dense top-layer, which was a common asymmetric morphology generated by a delaying liquid-liquid demixing mechanism [83]. Even though all membranes were stable in physiological settings, the modified membranes had better biocompatibility and antibacterial capabilities than the pristinePES membrane.

C. Nie et al. [84] used aramid nanofiber (ANF) to modify the surface of PES and PSF membranes via the blending method. The mixture of ANF and PES or PSF was prepared and forcefully agitated to make a homogeneous casting solution. The casting solution was de-gassed before being spin-coated on a glass plate and quickly soaked in distilled water to create the composite membranes. By improving FRR, suppressing protein adhesion, and inhibiting bacterial attachment, the modified membranes illustrated enhanced antifouling properties. Moreover, in comparison to the pristine membranes, platelet adsorption was reduced, plasma clotting time was enhanced, and coagulation and complement factor activation were repressed in the ANF-modified membranes, confirming improved hemocompatibility. The blending of ANF also improved the efficacy of the composite membranes in removing creatinine toxin.

ZW polymers are now being used to enhance the hemocompatibility of HD membranes, because of their outstanding antifouling qualities, decreased platelet adsorption, and activation. L. F. Fang [85] reported polyvinyl chloride (PVC) membrane modification with a novel zwitterionic polymer, methacryloyloxyethylphosphorylcholine-co-poly(propylene glycol) methacrylate (MPC-PPGMA) as the blending additive via non-solvent induced phase separation method. The surface-segregated MPC groups give the blend membranes with superior surface hydrophilicity and anti-biofouling properties because of the excellent hydration capacity of the zwitterionic part. The FRR for the blend membrane with 1 wt% MPC-PPGMA copolymer in the casting solution was greater than 97%. The stability and endurance of the copolymers in blend membranes, as well as the sustained antifouling property on modified membrane surfaces, were demonstrated by multi-cycle filtration and long-term stability testing. Through blending, S. Y. Choi et al. [86] created some novel oligomeric polyurethanes using methylene diphenyl diisocyanate (MDI) and sulfobetaine (SB), tetra-fluoro ethylene (TF), or SB/TF (SBTF). The surface migrating oligomers (SMOs) were synthesized from polyurethane backbones containing ZWs and fluorine and then blended with PVC to provide enhanced anti-thrombotic properties through the anti-fouling effect. Comparing the SMO (SBTFPU) combining both SB and hydrophobic TF parts was highly successful in terms of fouling inhibition and anti-thrombogenicity. Table 1 compares the increased membrane performance achieved by blending various additive blocks.

### 2.4. Coating

Surface coating is a facile, cost-effective, and environmentally acceptable technique of surface modification in which the coating material(s) create a thin layer that adheres to the membrane through non-covalently forces (weak van der Waals or hydrogen forces) [87,88,89]. Coating techniques are divided into five categories: (I) coating followed by heat curing, (II) deposition from a glow discharge plasma, (III) coating of a hydrophilic thin layer by physical adsorption, (IV) coating with a monolayer using analogous or Langmuir–Blodgett techniques, and (V) extrusion or casting of a two-polymer mixture by simultaneous spinning using, for example, a triple orifice spinneret. In the last method, as the membrane polymer and the top coating layer were mixed with various solvents [90,91], the adhesion between them could be improved.

Noteworthy is the fact that this coating method is suitable for flat-sheet membranes, and it is difficult to modify hollow fiber membranes using this method. Hollow fiber membranes have more complicated morphologies and structures than flat-sheet membranes, which limits surface modification studies, particularly interior-surface modification of hollow fiber membranes [89]. While no chemical reaction occurs in this method, the modified membranes are relatively stable due to the electrostatic interactions between the membrane and the coating layer. In this process, dense coating layers are placed over the entire membrane surface, resulting in thick smooth membrane surfaces with better selectivity and lower flux [92]. However, the pore size of the modified membrane can not be adjusted in this method and some coatings layers decrease membrane pore size, limiting the membrane ability to selectively transport molecules of different sizes [93]. Membrane degradation during performance or cleaning processes, as well as additional processing stages that are time-consuming, are also disadvantages of this method [94]. Various coating methods include physical adsorption, layer by layer (LBL), self-assembled monolayers (SAMs), spin-coating, spray-coating, Langmuir-Blodgett monolayer, and casting or extrusion between two polymer solutions by simultaneous spinning [4,90,95,96].

Zwitterionic polymers have distinguished out from other biocompatible and antifouling-coated layers for their excellent biocompatibility and great antifouling property [97,98]. In 2016, A. Venault et al. [99] developed a unique ZW copolymer, zP(4VP-r-ODA), by polymerizing 4-vinylpyrrolidone and octadecyl acrylate as a coating layer to improve the hemocompatibility of polypropylene (PP) membranes using a simple and inexpensive self-assembling approach. Commercial PP membranes were modified using a thermal evaporation coating method in this study [99]. Membranes were first washed in ethanol while being treated with an ultrasonic cleaner. The membrane discs were then placed in separate wells and coated with different coating solutions. The solvents were allowed to evaporate, and the membranes were then dried. By calculating the weight difference per unit surface area between the self-assembled membrane and the virgin membrane, the coating densities were determined. The modified membranes demonstrated good hemocompatibility and the ability to resist the adsorption of BSA, lysozyme proteins, blood cells, and escherichia coli. A. Venault et al. [100] also prepared new zwitterions p(MAO-DMEA) (synthesized through poly(maleic anhydride-alt-1-octadecene) and N,N-dimethylenediamine reaction) and p(MAO-DMPA) (synthesized through poly(maleic anhydride-alt-1-octadecene reaction) and 3-(dimethylamino)1-propylamine) to modify the antifouling performance of poly(vinylidene fluoride) (PVDF) membranes via a self-assembled anchoring method. In this study, ZW copolymers were first prepared by ring-opening zwitterionization and then self-assembled by thermal evaporation coating onto PVDF membranes, which showed excellent antifouling ability in terms of preventing BSA and FN attachmentas well as repelling various blood cells, platelets, and bacteria.

Using a two-step spin-coating method, C. He. et al. [101] reported a novel and hemocompatible dual-layered PES membrane made up of a graphene oxide top layer and a sulfonated polyanion co-doped hydrogel bottom layer (GO-SPHF) by a two-step spin-coating method. To prepare the top casting solution, Acrylic acid (AA) and sodium styrene sulfonate (SSNa, 90%) were cross-linked via in situ free-radical copolymerizations in the PES solution in which GO was added to increase miscibility with PES through hydrophobic and π–π interaction. The liquid-liquid phase inversion method was used to obtain dual-layered membranes by spinning pristine PES solution onto glass plates in two steps. The two layers of the dual-layered functionalized membranes had a distinct border that was effectively combined (Figure 3a). The modified membranes showed good durability for a variety of applications while exhibiting minimal tensile strength loss. The modified membranes exhibited outstanding anticoagulant characteristics, low inflammation, decreased platelet adsorption, and excellent cytocompatibility, giving them exceptional hemocompatibility for a variety of blood-contact applications. Furthermore, they retained high durability for a variety of applications while exhibiting minimal tensile strength loss.

B. R. Knowles et al. [102] reported silica nanoparticles modified with sulfobetaine siloxane as a novel and efficient fouling resistance layer for surface functionalization. To create hydrophilic antifouling coatings, silica nanoparticle suspensions were designed with SB at different pH levels and coated as thin films by a simple spin-coating procedure (Figure 3b). Surface modification of pre-deposited SiNPs functionalized with SB offers a quick technique of introducing hydrophilic chemistries to surfaces with strong resistance to protein adhesion and may be used to create robust antifouling coatings appropriate for long-term applications.

S. H. Chen et al. [103] used free radical polymerization and RAFT polymerization to produce a set of random and block zwitterion copolymers that were used for zwitterionization of PP membranes using a-convenient coating process. A pre-wetting treatment can adjust the chain orientation of the above-mentioned copolymers that are bonded to hydrophobic PP membranes, leading to enhanced antifouling and hemocompatibility properties. The resultant modified membranes showed superior bio-inert characteristics, excellent hemocompatibility in human whole blood, significant antifouling capacity against leukocyte and thrombocyte adhesion, as well as different proteins and platelet attachment.

X. Lin et al. [17] reported a new functional amphiphilic ZW copolymer coated on a PVC surface that was constituted of both super hydrophobic carboxybetaine acrylamide (CBAA) units and hydrophobic/photosensitive N-(4-benzoylphenyl) acrylamide (BPAA) units. Surface modification was performed using hemocompatible polymers containing carboxybetaine (CB) parts and photosensitive cross-linking moieties, accompanied by UV light illumination. The modified PVC materials showed superior resistance to protein attachment, platelet adsorption, and complement activation in real circumstances.

Spin coating is a versatile procedure that involves either casting the solution onto rotating support or casting it onto static support and then spinning it [104,105,106]. When compared to immersion assembly, spin assembly produces more homogeneous, thinner, and smoother films in a quicker time frame, making it more appealing for membrane design and development [107]. Z. Zhou et al. [108] reported a novel method of spin-coating assisted interfacial polymerization (SCIP) to fabricate ultrathin polyamide (PA) membranes for nano-filtration (Figure 4). The spin coating produces a homogeneous diamine distribution that is mostly localized outside of the surface pores instead of inside the substrate. By forming ultrathin PA nano-films smaller than 10 nm and minimizing the size of back surface protuberances, improvement of water permeability is achieved as preserving mechanical stability.

Ren et al. [109] designed a durable protein-resistant multilayer system using poly(sulfobetaine methacrylate) (PSBMA) and tannic acid (TA) through an LBL assembly approach to improving hydrophilicity and resistance to proteins. Increasing the number of bilayers in the (TA/PSBMA)n multilayers resulted in a change in surface hydrophilicity, and (TA/PSBMA)20 multilayers demonstrated superior hydrophilicity and antifouling properties than others with fewer bilayer numbers (*n* = 5, 10, 15).

H.W. Chen et al. [110] developed an amine-rich surface for the conjugation of ZW polymers using LBL polyelectrolyte deposition. In this method, an ultra-thin film is built by alternating positively and negatively charged polyelectrolytes adsorbing on a substrate. Polyzwitterion with a carboxylated end was first prepared with different chain lengths before being conjugated onto TLP-coated substrates. Cell adsorption and protein attachment were almost prohibited on the polyzwitterion-modified surfaces.

Q. Chen et al. [111] developed new biocompatible and cytocompatible PES membranes using an HB-LBL assembly of anti-oxidative TA and hemocompatible poly (N-acryloyl morpholine) (PACMO). The PES membrane was alternately immersed in TA and PACMO solutions to create the multilayers. The pristine PES membrane was first placed in the TA aqueous solution for 20 min and then washed three times with distilled water. The membrane was then soaked in PACMO aqueous solution and again washed with distilled water. Repeated both coating and washing operations yielded the desired number of bilayers as shown in Figure 5a. Because TA is a powerful natural antioxidant, TA-PACMO functionalized membranes are resistant to oxidative stress. PACMO also boosted hydrophilicity and hemocompatibility, which improved anticoagulant activity and reduced platelet and red blood cell adherence on membrane surfaces, as well as lowered the likelihood of complications.

Even though electrostatic LBL cannot be utilized to create multilayers of zwitterionic polymers due to the inner salt structure, Xie et al. [112] used Schiff-based LBL assembly to create an antifouling PES membrane surface (Figure 5b). In the first step, amino-abundant ZW polymers of PEI-SBMA and aldehydes-rich oxidized sodium alginate (OSA) were prepared; then, a fouling resistance coating was produced by assembling the two macromolecules using the facile Schiff-based LBL assembly with the PEI-SBMA layer remaining as the out-layer for all the samples. Following that, the synthesized membranes (PES/PEI-SBMA/OSA-n) were soaked in a silver nitrate solution and sodium borohydride, respectively, to create Ag-loaded membranes (PES/PEI-SBMA-Ag and PES/PEI-SBMA/OSA-n-Ag). The loaded Ag NPs membranes could prevent bacterial growth, and bactericidal action could be improved by raising layer numbers to increase the loaded Ag NPs ability. The results demonstrated that PES/PEI-SBMA-Ag and PES/PEI-SBMA/OSA-n-Ag membranes could be useful in clinical bacterial contamination control for a variety of biomedical implants.

A notable recent invention is ‘Mussel’ inspired chemistry, in which the membrane is coated with dopamine analogs. By adjusting temperature, deposition time, pH, and atmosphere, the deposition method can be easily controlled. High surface hydrophilicity and antifouling qualities are provided by polydopamine (PDA) coating. PDA is also capable of interacting with many different molecules, making it a useful platform for applying covalently grafted functional layers to substrates [113,114,115,116]. These membranes, however, are often expensive and have low chemical stability, especially in an alkaline environment mostly, which limits their use on a large scale [117,118,119].

R. Zhou et al. [120] reported the modification of polypropylene membranes with PSBM through a facile and efficient co-deposition procedure in one step. Co-deposition is simply achieved by immersing microporous polypropylene membrane (MPPMs) samples in a one-pot mixture of dopamine alkaline solution and varying concentrations of PSBMA as illustrated in Figure 6a. The average pore size of the surfaces of the unmodified membrane is around 0.20 m, according to FESEM images, and there was no substantial alteration for the membrane deposited with 1:0 PSBMA/DPA. Because the membrane surfaces are covered in a thin layer of PDA, the pore size appears to shrink slightly following PDA deposition. When the membrane is co-deposited with PSBMA/PDA, this phenomenon is amplified. The modified membranes showed better hydrophilicity, lower water flux reduction, and higher water flux recovery than pure membranes.

A straightforward and efficient method to produce superior biofouling resistance coatings with good stability has been proposed by L. Yao [121] based on the co-deposition of PDA and an amino-enriched ZW polymer. PDA and polyethyleneimine-quaternized derivative (PEI-S) were deposited on the PES membrane surface in water at room temperature (Figure 6b). The addition of PEI-S to PES membranes could significantly increase their antifouling capacity. Moreover, because of covalent cross-linking, the chemical durability of the co-deposited polymers (M-PDA/PEI-S) was greatly improved over that of the PDA-coated membrane (M-PDA). Importantly, the M-PDA/PEI-S showed exceptional stability under extreme alkaline conditions due to the cation-interaction (Figure 6c).

Zhang et al. [122] developed a facile and sustainable approach to immobilizing a heparin-like coating consisting of TA, DAS, and SMP on a PES ultrafiltration membrane through the hydrophobic interactions and hydrogen bonding of TA with the PES membrane, the cross-linking of D-asparagine (DAS) with TA, and a Michael addition reaction of sodium 3-mercapto-1-propane sulfonic acid (SMP) with TA (Figure 6d). The insertion of hydrophilic functional groups as –COOH and –SO_3_ to the membrane surface resulted in a heparin mimetic coating which increased the hemocompatibility of the modified membrane. Different membranes with high flux and excellent antifouling capabilities were created by adjusting the coating conditions. The TA/DASSMP-modified coating did not affect the membrane pores of the ultrafiltration membrane, according to SEM and AFM pictures, and the modified membrane PEST/D-S had a homogeneous surface. A summary of some studied surface-coating modifications is given in Table 2.

## 3. Chemical Modification

The chemical modification involves changing the surface of membranes by using chemical reactions to modify molecules with various functional groups [13,91,123] that are then cross-linked or immobilized with biocompatible compounds. The surface chemistry of the membranes will certainly be altered as a result of chemical modifications. It provides long-term stability and can be applied to both polymers and inorganic membrane materials. A variety of chemical techniques are available, including chemical grafting, UV-induced grafting, controlled radical polymerization, click chemistry, mussel-inspired chemistry, initiated chemical vapor deposition (iCVD), plasma techniques, and enzymatic treatment. Modification of membrane surfaces chemically can provide desired surface properties while retaining the bulk polymers desired mechanical properties, chemical resistance, and morphology.

However, the modified membranes in this method frequently exhibit non-uniform surfaces, and the drastic modification approach often affects their characteristics. A long-term modification may cause the bulk materials to be etched. Furthermore, it is crucial to carefully control the modification time of harsh modification procedures, such as plasma, high-energy radiation, and ozone treatment.

### 3.1. Grafting

The surface of membranes naturally possesses functional groups such as carboxylic acids and primary amines that can be utilized to add various functional groups. Chemical grafting may produce free radicals and ionic species, initiating polymerization that serves as a driving force for bonding of the modification compounds. ZW species, polyethylene glycol, polydopamine, as well as inorganic nanoparticles are some compounds that have been used to prepare antifouling membranes [26,124,125,126,127]. Surface grafting methods are attracting a lot of research attention among other techniques because of their flexibility in tailoring desired surface features with different monomers and accuracy in imparting grafts at specified locations onto the membrane. Additionally, surface grating provides a facile and controllable method to incorporate polymer compounds with large surface areas and can improve membrane chemical stability and performance during hemodialysis therapy.

G.V. Dizon [128] developed an effective and novel “grafting-to” method for surface zwitterionization of polydimethylsiloxane (PDMS) to increase membrane antifouling and hemocompatibility. They used tannic acid Fe(III) as a first coating layer on PDMS to form numerous -OH groups that could be exploited in the ring-opening reaction of the GMA parts of poly(glycidyl methacrylate-co-sulfobetaine methacrylate) (PGMA-co-SBMA)copolymer (Figure 7a). The intrinsic key characteristics of PDMS, including physical and mechanical characteristics and optical transparency, are preserved using this approach. As a result of the pre-treatment procedure with the polyphenyl-metal complex, a layer with many –OH groups are formed that is capable of allowing the covalent attachment of ZW compounds. When in contact with a range of biomolecules, including escherichia coli, erythrocytes, thrombocytes, plasma proteins, and human blood, the as-prepared PDMS displayed outstanding biocompatibility with no changes in mechanical properties or optical transparency.

“Grafting from” refers to grafting in which the initiators are connected to the surface before polymerization, leading to better density control. As a result of this method, surface-anchored polymers are usually poorly controlled in terms of polymer structure, but they may be able to reach a variety of chain lengths and grafting densities under comparatively straightforward reaction conditions. However, this strategy is constrained by a multi-step procedure that makes it difficult and time-consuming. To address this problem, P. Weng et al. [129] designed a one-step approach for cellulose membrane surface zwitterionization based on alkoxysilane polycondensation to enhance its antifouling property and biocompatibility. Three alkoxysilane coupling agents with pendant ZWs were produced and grafted onto the cellulose membrane (CM) surface. Protein adhesion, platelet adhesion, and cell attachment resistance were all outstanding in the zwitterions-grafted membranes. The modified membranes also showed nearly the same permeability as the pristine membranes. E. N. Simsek et al. [130] used bulk modification via nitration and reduction processes to add amino or amide (acetyl), functional groups, to the aromatic ring of PES, resulting in improved hydrophilicity and fouling resistance characteristics. Initially, a nitration reaction was used to attach NO_2_ groups to the aromatic ring of PES, followed by reduction processes and acetylation of the amine groups with acetic anhydride to produce PES-NHAc (Figure 7b). Zhang et al. [131] also used the grafting process as a cheap and reliable approach for attaching ZWs groups to the backbone of PES to create two modified PES membranes (PES-CB & PES-SB) with superiorhydrophilicity and antifouling capabilities. The prepared membranes showed a 95 percent recovery ratio of the lysozyme solution flux, whereas the virgin membranes only had a 25% recovery ratio.

Radiation-induced graft copolymerization (RIGC) is a flexible approach for covalently bonding preferential antifouling substituents to a manageable amount of grafting groups with the use of less dangerous chemicals and potentially improving fouling issues [132]. It can be conducted using low-energy (UV and plasma) and high-energy (γ-rays) and electron beam (EB) radiation. J. F. Jhong et al. [133] used plasma-induced RIGC to graft PSBMA and hydrophilic poly(ethylene glycol) methacrylate (PEGMA) onto poly(tetrafluoroethylene) (ePTFE) membranes. The wettability of the ePTFE-g-poly(PSBMA) membrane was high, and it became less adherent to protein, tissue cells, and bacteria. The use of plasma treatment to produce membranes with high hemocompatibility, biocompatibility, and minimal biofouling by RIGC with ZW monomers could be highly advantageous for HD treatment.

J. Zhao et al. [134] reported O_2_ plasma pretreatment and UV-irradiated technique to graft a ZW polymer, [3-(methacryloylamino)propyl] dimethyl(3-sulfopropyl) ammonium hydroxide (MPDSAH), onto a polypropylene non-woven fabric (NWF) membrane (Figure 7c). Fourier transforms infrared spectroscopy (FTIR-ATR), static WCA measurement, and X-ray photoelectron spectroscopy (XPS) were used to analyze the surfaces of the modified NWF membranes. The WCA for the membranes decreased from 123° to 17° when the grafting density of poly(MPDSAH) was raised from 0 to 349.2 g/cm^2^, whereas equilibrium water adsorption achieved a peak at the grafting density of 120.5 g/cm^2^. Reduced protein attachment and platelet adsorption on the modified membrane demonstrated improved antifouling properties due to the addition of zwitterionic polymers. Plasma-induced is a rapid technique for surface modification that produces clean and uniform grafts on the membrane surface by comprising four fundamental effects like as cleaning, cross-linking, ablation, and chemical alteration [135]. The change in surface energy caused by plasma-induced grafting has also a significant impact on membrane fouling [136].

**Figure 7 membranes-12-01063-f007:**
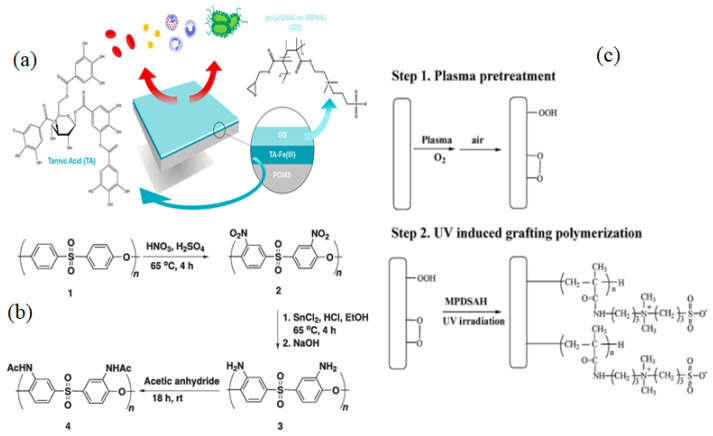
(**a**) Grafting zwitterion on PDMS surface [128]. (**b**) Schematically reaction of nitration, reduction, and acetylation on PES membrane [130], (**c**) Synthetic procedure of NWF-poly(MPDSAH) membrane [134].

Y. P. Tang et al. [137] reported using a zwitterionic graft copolymerization technique to create an antifouling lumen surface on a PVDF hollow fiber membrane. Because the inside-out filtering mode offers proper control over module hydrodynamics, PVDF hollow fiber membranes with antifouling lumen surfaces are of tremendous interest and importance for several industrial uses. This research group was the first to use thermal-induced graft copolymerization to fix a ZW polymer on the surface of a PVDF hollow fiber membrane, to make the antifouling PVDF membrane. Two phases are involved in the graft copolymerization process: ozone-treated surface pre-activation and grafting of SBMA as illustrated in Figure 8a. By modifying process conditions such as ozone treatment and grafting time, the grafting density and chain length were systematically tuned. Ozone-assisted oxidation or grafting can provide spread grafts or peroxides on the surface of the membrane even with a complex configuration. Modified membranes illustrated notable fouling inhibition when filtered through a 2.0 wt% solution of BSA. Pretreating PVDF membranes with ozone is an efficient and simple way to change the surface of commonly available membranes without affecting their remarkable physical and chemical properties [138,139].

Y. N. Chou [140] used the ZW copolymer formulation of poly(glycidyl methacrylate-co-sulfobetaine methacrylate) (poly(GMA-co-SBMA) to develop a simple, practical, and cost-effective grafting process that forms a stable chemisorption layer on a range of surfaces. The ring-opening process between the epoxide parts of GMA and nucleophilic -OH groups is enough reactive for grafting poly(GMA-co-SBMA). After the surface is pre-assisted with UV and ozone, the poly(GMAco-SBMA) can bond covalently with a variety of surfaces, such as ceramic, polymer, and metal substrates owing to the -OH groups (Figure 8b). In this study [140], an SB formulation and grafting method are proposed to support effective zwitterionic layer formation on various membrane surfaces with nonspecific fouling inhibition for general bioadhesive constituents, like plasma proteins, tissue cells, platelets, erythrocytes, and bacteria. Y. Xia et al. [141] described grafting of poly(sodium p-styrene sulfonate -co- sulfobetaine methacrylate) P(SSNa-co-SBMA) to PES membranes using ammonium persulfate (APS) to produce -OH groups on the membrane surfaces for the polymerization process. In situ cross-linking polymerization was applied to create –OH groups upon the PES membrane, followed by free radical polymerization initiated by APS to generate functional polymers of P(SSNa-co-SBMA) (Figure 8c). Results showed that the membranes with good blood compatibility and antifouling properties were prepared by using a reliable grafting-from technique, which was a convenient and efficient method of modifying membranes. Furthermore, the modification did not affect membrane permeability. In another study, using in situ cross-linking, C. Wang et al. [142] prepared carboxylic PES composite membranes and then post-functionalized them with sodium styrene sulfonate (SSNa) through a “graft from” method. PSSNa was grafted on the membrane surfaces after liquid-liquid phase inversion to improve the hemocompatibility and fouling resistance of the modified membrane. It should be feasible to introduce functional groups to membrane surfaces via this facile, versatile crosslinking technique, and the post-functionalization operation could endow the membranes with desired qualities.

In summary, surface grating techniques have gained significant attention in recent years due to their capability in achieving desired properties through various grafting monomers as well as their precision in implanting grafts at precisely the locations required. In addition to these benefits, surface grating can considerably improve membrane chemical stability which is critical for improving membrane hemocompatibility and anti-fouling capabilities. Although each of the surface grafting methods has advantages and disadvantages, there are also different factors that affect their efficiency. Bulk modification, as an old-fashioned modification technique, has a stable modification impact, but it is an unsuccessful method due to the hazardous organic solvents, harsh environment, and time-consuming processes [89,143,144]. Surface grafting is a straightforward, useful, and reliable technique for membrane modification that has been frequently used. Due to the covalent attachment, the surface of membranes might be modified with a variety of features by using various grafting agents with high densities and long-lasting chemical stability. Free radicals are typically created and transported to the membrane to commence polymerization and produce grafting copolymers in chemical grafting. The use of a chemical activator or cleavage agent to induce thermally initiated polymerization is a straightforward technique in membrane modification. Though surface grafting has a durable modifying effect, obtaining active sites usually necessitates sophisticated physical or chemical treatments, which restricts its use on a broad scale [145]. Furthermore, some investigations have shown that the pore structure might vary as a function of the grafting procedure, either enlarging or shrinking [146]. PIG, as a type of grafting procedure, is a simple, cheap, safe, and adaptable approach that is well-suited to MBR technology. Without any chemical or physical pre-treatment, the membrane polymers can be treated with photo-induced grafting (PIG) in their natural condition, minimizing time and resources. At the interface between the monomer and the membrane polymer, the PIG process can also creat desirable and localized active species [147]. Polymer grafting can also be done at a low temperature under mild operation circumstances, making it a safe synthesis technique [148]. It is also a versatile process that can be adjusted and combined with other techniques. However, the majority of these studies are still conducted on a small scale in batch tests and have not been thoroughly used in the MBR system. It should be mentioned that PIG is less effective on hollow fiber membranes since UV irradiation may not reach the membrane inside the surface [148,149]. Additionally, severe UV irradiation might cause irreparable damage to the polymer backbone in some cases. Similar to PIG, plasma treatment and plasma-induced grafting have drawbacks such as the complexity of scaling up to a continuous process and limited application research on MBR configurations. Furthermore, surface plasma and irradiation procedures may have an impact on membrane structure and mechanical characteristics. Various variables like voltage, oxidation time, monomer content, and hydroperoxide concentration must be controlled to optimize ozone-induced grafting, making it a difficult and time-consuming procedure.

Overall, a wide range of grafting procedures have been investigated, but comparing prior and current studies is challenging because of the non-uniformity of results for the various techniques. A variety of studies have been conducted to characterize WCA, surface roughness, and hydrophilicity, while others have looked at aspects such as water flux, FRR, relative flux ratio, platelet, and protein fouling. Table 3 summarizes a comparison of the various grafting procedures outlined in Section 3.1.

#### 3.1.1. Radical Polymerization, Initiated Vapor Deposition (iCVD) & Click Chemistry

Chemical grafting can also be carried out by using free radicals and ionic species to initiate polymerization, which is responsible for attaching the modification layer. Free radical polymerization, controlled radical polymerization, initiated chemical vapor deposition (iCVD), and click chemistry are some common methods for the grafting of zwitterion-containing units to natural polymers [22,26].

P. Saha et al. [150] developed a zwitterionic poly(phosphobetaine) (PMPC) microgel with excellent antifouling capabilities by thiol-epoxy click reactions mediated by macroreversible addition-fragmentation chain transfer (macro-RAFT). The effectiveness of the “grafting to” approach is demonstrated by the introduction of zwitterionic PMPC polymers with suitable chains and narrow distribution onto a poly(N-vinylcaprolactam-co-glycidyl methacrylate) (PVG) copolymer matrix.

In a subsequent step, PEG-based were utilized as effective cross-linkers to produce microgels with enhanced swelling and antifouling abilities. The fouling resistance of microgels was significantly influenced by the molecular weight of PMPC and PEG-NH_2_ parts. In a similar study, Z. Nadizadeh et al. [151] also used surface-initiated reversible addition-fragmentation chain transfer (SI-RAFT) polymerization as a promising radical grafting technique to uniformly modify nanofiltration (NF) membranes with functional groups while avoiding nonspecific protein fouling. Polyamide thin-film composite NF membranes were produced by interfacial polymerization between trimesoyl chloride (TMC) and a combination of diamines (i.e., 1,3-phenylenediamine (MPD) and (3,5-diamino phenyl) methanol (DAPM). The reaction of the polyamide thin film composite layer with α-bromoisobutyryl bromide (BIBB) was then performed. The amount of BIBB on the surface of the produced membranes could be effectively increased by utilizing the DAPM treatment, based on their result. Lastly, a ZW polymer, poly [(2-methacryloyloxy)ethyl]dimethyl [3-sulfopropyl]ammonium hydroxide (pMEDSAH), was effectively grafted on the membrane surface by the RAFT polymerization process. The amount of nonspecific protein attachment on the produced pMEDSAH-grafted membrane was reduced to >96 percent. This polyamide TFC modification technique offers a simple and efficient method for stabilizing the RAFT initiator to produce the non-fouling separation membrane via interfacial polymerization.

The synthesis of fouling resistance PES membranes modified with zwitterionic polymers surface grafted from a reactive amphiphilic copolymer additive was presented by Y. F. Zhao [152]. PES-b-PHEMA (amphiphilic polyethersulfone-block-poly(2-hydroxyethyl methacrylate)) was produced at first and employed as a blending additive in phase inversion PES membranes. The surface-loaded PHEMA blocks on the membrane surface served as anchoring to keep the starting site immobilized. SI-ATRP was used to graft PSBMA onto the PES membranes (Figure 9a). By altering the modification time, the PSBMA’s grafting yield may be controlled. When compared to the original membrane, the modified PES membrane showed improved hydrophilicity, excellent antifouling performance, and high blood compatibility (protein attachment, platelet adsorption, and plasma recalcification time).

Using iCVD and di-tert-butyl peroxide as a radical initiator, M.N. Subramaniam [153] modified PVDF hollow fiber membranes with thin layers of hydroxypropyl methacrylate (HPMA) (Figure 9b). This technique was able to thoroughly coat PVDF membranes, resulting in a homogeneous deposition of hydrophilic HPMA on the membrane surface of 50 nm and 100 nm thickness. The HPMA-covered membranes had dramatically increased surface hydrophilicity, as evidenced by a decrease in WCA from 78.1° to 23.6°, while the thin-film coating lessened membrane surface roughness. The membrane permeability and rejection of 100HPMA-PVDF membranes were increased by 50.8 L/m^2^ h and 83.1%, respectively. Owing to its considerably increased surface hydrophilicity and decreased surface roughness, the 100HPMA-PVDF membrane showed consistently high permeation and rejection performances after four filtration runs.

T. Xiang et al. [127] reported using ATRP and click chemistry to incorporate ZW polymers of PSBMA, negatively charged polymers of poly(sodium methacrylate) (PNaMAA), and/or poly(sodium p-styrene sulfonate) (PNaSS) into PSF membranes to enhance protein resistance and hemocompatibility of PSF membrane (Figure 9c). Because click chemistry is a facile and one-step process to quickly and reliably binding materials together, the combination of ATRP and click chemistry outperformed other techniques such as blending, coating, and surface-initiated ATRP. The modified membranes have good antifouling and bacterial adhesion properties, as well as increased blood compatibility, particularly anticoagulant properties. S. Zheng et al. [154] reported an anti-fouling silicon surface using a zwitterionic polymer, polySBMA by clicking chemistry. The following three steps were used to functionalize the silicon surface with polySBMA: (1) RAFT polymerization was used to make an alkyne-terminated polySBMA; (2) a self-assembled monolayer with bromine end groups was formed on the silicon surface, and then the bromine functional groups were modified with azide parts; and (3) the polySBMA was bonded to the silicon surface via an azide–alkyne cycloaddition click reaction. For both BSA protein and bacterial cells, the polySBMA-modified silicon surface was proved to possess anti-nonspecific adsorption capabilities. The polySBMA-modified silicon surface demonstrated better biocompatibility and powerful antifouling properties in whole-blood adsorption studies.

In conclusion, the iCVD technique and radical polymerization have also been widely employed for grafting ZW monomers onto membrane surfaces using an appropriate initiator. Unlike uncontrolled radical polymerizations, which lead to radical recombination and disproportionation, living radical polymerization allows enhanced control over molecular weight as well as three-dimensional architecture. Chemical modification methods RAFT and ATRP are popular because of the immobilized initiator on the surface that makes it possible to generate brush patterns on the membrane with precision [151,153,155,156,157,158,159]. The graft ratio of ZW polymers on membrane surfaces is determined by the number of initiators immobilized on membrane surfaces, whereas the degree of polymerization is determined by ATRP reaction time [160,161,162]. Furthermore, despite harmful organic solvents, these controlled procedures are compatible with aqueous media, making them environmentally friendly processes. The ATRP method provides a benign reaction condition and fewer operational stages, and most chemicals are readily available [163]. ATRP is one of the most powerful ways of functionalizing membrane surfaces with ZWs polymers. However, in this process, copper ion removal is still challenging. Although RAFT polymerization rate is slower than ATRP’s, it is suited for a broad range of monomers and can control polymerization without using metal catalysts [164,165,166]. To achieve active polymerization, RAFT uses standard initiators like AIBN to invoke radicals, and the chain transfer agent 4-cyanopentanoic acid dithiocarbamate (CPADB) to accomplish the reaction [167,168]. The membranes modified by living/controlled radical polymerization show stable antifouling properties as well as the controlled introduction of high-density graft chains. The complex modification techniques, on the other hand, inhibit large-scale application. Furthermore, due to membrane pore blocking produced by grafted zwitterionic polymers, most surface grafting procedures result in a decrease in permeation flux [169]. The grafting content is a critical factor in “surface grafting” that is directly connected to membrane permeability and antifouling effectiveness and should be taken into account. Almost all studies indicate that the antifouling capabilities of membranes improve with increasing ZW grafting density due to a thicker hydration shell and increased strict hindrance, and then remain nearly unchanged after the grafting density reaches a specific level. However, as the ZW grafting density grows, the permeate flux increases at first and then declines, which can be described by improved hydrophilic efficiency, clogging, and shrinking of membrane pores [26,170]. Consequently, permeability and antifouling are competing goals in all types of grafting methods. Furthermore, increasing grafting density causes a decline in surface roughness [171,172], which may be beneficial to antifouling. So when ZW grafting density is raised, it is difficult to tell whether the improvement in antifouling qualities is due to a reduction in surface roughness or an enhancement in hydrophilicity.

Using radical polymerization (e.g., ATRP) in combination with click chemistry represents a big step forward in modification chemistry, as the initiator site on the membrane can be employed as a reaction group for several copper-catalyzed reactions. When the membrane surface is modified with click chemistry, multiple reaction sites are provided on the membrane surface and ATRP enables control of the process. This method permits the grafting of polymers with various conformations and structures onto membranes with specific compositions. Click chemistry has been applied to many membrane materials and provides an effective technique for membrane surface modification (e.g., PES [173,174], PSF [175,176]). The potential of click chemistry to give greater site selectivity and essentially quantitative transformation under mild conditions, with almost no reactions or by-products, is a significant benefit [177]. The one-step click chemistry reaction holds significant potential for cost-effective and long-term surface modification. Overall, click chemistry is a grafting technique that has many of the benefits and drawbacks of other grafting techniques.

#### 3.1.2. Mussel-Inspired Chemistry

Researchers have been interested in the mussel-inspired hydrophilic modification strategy because of the moderate experiment settings, high modification efficacy, universality, and robust bio-adhesion of marine mussels [178,179]. PDA and heparin are well-known “bio-glues” that adhere to a variety of materials and surfaces. In recent years, much research has gone into studying the mussel adhesion mechanism and expanding its applicability in the production of hemocompatible membranes. The method is simple and easy to control by changing the temperature, pH, deposition length, and environment. Dopamine and its derivatives have been shown to be useful, facile, and stable adhesives for anchoring ZW polymers onto membrane surfaces, providing strong antifouling capabilities to the membranes [180,181]. Biomimetic adhesives are typically useful to make or modify dense membranes because they cover the entire surface with a long deposition period, sealing and shrinking membrane pores while also reducing surface roughness. One of the most extensively used ways for attaching a ZW layer to the membrane surface via thiol and amine chemistry is the employment of a mussel-inspired dopamine adhesive functional layer. According to recent research, there are four main methods to dopamine and zwitterion conjugation [182]: (1) conjugation of dopamine with ZW via direct modification of ZWs with the dopamine functional moiety; (2) codeposition of dopamine with ZW polymers; (3) zwitterionic post modification of the PDA coated surface; and (4) surface-initiated polymerization of zwitterionic polymers using dopamin modified initiators (Figure 10).

The simplicity of directly attaching ZW polymers with adhesive layers to the surface in one step to produce an antifouling surface coating with improved packing density is particularly appealing. Because zwitterionic polymers are very water-soluble, a robust surface binding group on the polymer is required to promote increased packing density and stability. Gao et al. [183] synthesized two catechol groups including a polymer (pCB2catechol2) with two zwitterionic poly(carboxy betaine) (pCB) arms to increase the grafting density of catechol-containing polymer. pCB2–catechol2 had the strongest non-fouling capabilities on the grafted surface, with robust surface binding and improved surface coverage, due to the increased surface anchoring group DOPA in the polymer chain (Figure 11a). Dopamine-assisted co-deposition simplifies zwitterionic polymer grafting and enables a coating to attain maximum functionality by simultaneously depositing and functionalizing in a single step. This method is more resilient, and flexible, and can be easily modified by modifying the pH, deposition duration, concentration, and environment than the usual ‘grafting to’ approach. W. Xu et al. [184] studied PVC modification by grafting zwitterion polymer (PESX) after co-deposition treatment of PDA and PESX with an optimum feeding ratio, resulting in PDA/PESX-PESX coatings. According to the results, the modified membrane prevents blood components and bacteria from adhering to the surface and maintains high hemocompatibility and cytocompatibility. In another study, Y. Xie et al. [185] used a co-deposition of PDA and zwitterionic polymer, followed by the incorporation of bactericidal silver nanoparticles, to create a unique mussel-inspired antibacterial and non-fouling PES membrane (Ag-NPs). To create an antifouling surface, polyethyleneimine-graft-sulfobetaine methacrylate (PEI-SBMA) was first crosslinked with PDA and co-deposited onto the PES membrane surface. The Ag-NPs were then synthesized in situ on the membrane surface without the use of any external reducing agents, making mussel-inspired antifouling and antibacterial membranes simple to create.

It is more advantageous to use the catechol motif of PDA in post-modification strategies, especially when conjugating it with thiol or amine-containing molecules. N. Shahkaramipour et al. [71] modified the PES membrane with the newly synthesized zwitterionic polymer using a direct attachment approach. They produced copolymers out of hydrophilic zwitterionic phosphobetaine methacrylate (MPC) and thiol functional groups, the latter of which was used to react with catechol and covalently graft the ZW polymers onto the PES membrane surface via the Michael addition reaction (Figure 11b). Because of the PDA layer, the surface coating reduced pore size and porosity, which decreased water permeance. The surface coating, on the other hand, became more hydrophilic as a result of ZW polymers, as it was illustrated by the decrease in WCA. The modified membranes were able to be used for long periods due to covalent bonding between PDA and water-soluble ZW polymers, which is a major concern in the co-deposition technique, which uses noncovalent linkages to form thin films of PDA and zwitterionic polymers on the membrane surface.

Grafting ZW monomers from a surface via SI-ATRP is among the most powerful ways to avoid biofouling. This SI-ATRP has been extensively applied to produce fouling-resistance polymer brushes with high packing densities and adjustable film thicknesses owing to its controlled/living characteristic [182,186]. The combination of SI-ATRP, which is used for selective substrate surfaces, with powerful adhesive dopamine chemistry has been widely investigated to deposit the PDA layer on a variety of surfaces and to serve as a platform for subsequently immobilizing ATRP initiators [187,188]. Catecholamine-initiated ATRP can be applied to practically any substrate, regardless of surface chemical properties, forms, or sizes, providing unique, powerful, and versatile methods for functionalizing a wide range of materials [46,189,190]. PVDF membranes with enhanced antifouling properties and hemocompatibility were reported by Jiang et al. [191] by using polydopamine-mediated ATRP. In the first step, PDA was initially synthesized by oxidation and self-polymerization under basic conditions. PDA was used as an additive to prepare PVDF membranes via NIPS. Then, using ATRP, a commonly used zwitterionic polymer, PSBMA was effectively grafted from the trapped PDA in the membrane (Figure 11c). Because of the chemical reactivity of PDA and its robust interactions with a wide range of solid substrates, this method represents a versatile strategy for hydrophilic and biocompatible modification of hydrophobic polymer membranes. D. M. Davenport et al. [192] also used surface SI-ATRP and PDA to graft PSBMA brushes to the PVDF membrane surface. To immobilize bromine initiator groups to UF membrane surfaces, researchers used PDA and SI-ATRP with varying reaction periods to graft PSBMA brushes of various lengths. N. Li et al. [193] recently published a universal technique for making stable zwitterionic polymer brushes that combines the benefits of both PDA chemistry and ARGET-ATRP. In this study, the PDA layer was first dip-coated on a substrate, then 3-trimethoxysilyl propyl 2-bromo-2-methylpropionate (SiBr, ATRP initiator) was covalently immobilized on the PDA-coated surface via a condensation process between the silicon hydroxyl and the PDA hydroxyl groups. Next, without deoxygenation, SI-ARGET-ATRP was carried out in a zwitterionic monomer solution catalyzed by CuBr_2_. The stability, good antifouling, and excellent blood compatibility of these easily produced zwitterionic polymer brush coatings were proven.

**Figure 11 membranes-12-01063-f011:**
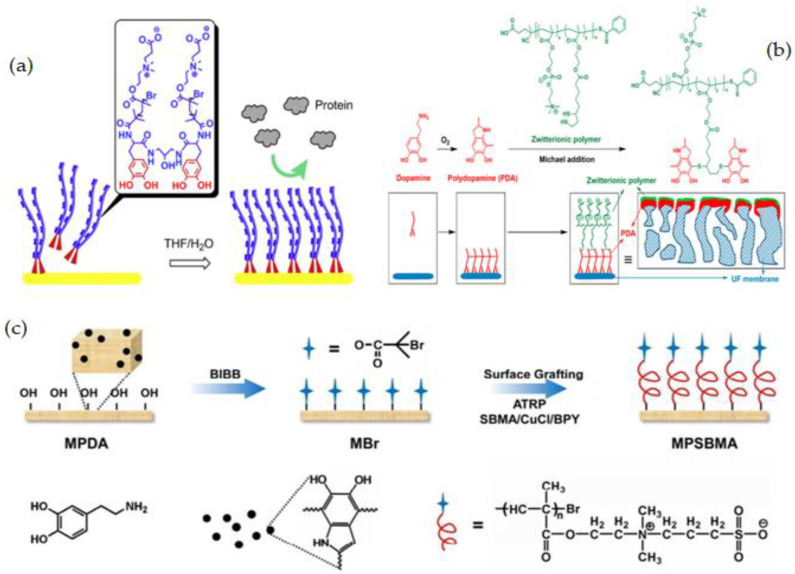
(**a**) Au surface grafted with pCB2-catechol2-Direct modification method [183], (**b**) Schematically preparation of PES modified ZWs using PDA as a Bio-glue intermediate layer [71], (**c**) Grafting of PSBMA zwitterions onto PVDF/PDA blend membrane via PDA-ATRP [191].

Heparin is well-known for its anticoagulant qualities, and its primary application for surface modification of polymers is to enhance biocompatibility, owing to the anti-adhesive qualities of heparinized surfaces toward platelets. In membrane modification, heparin is usually immobilized by surface covalent bonding and polymerization or by interaction with surface ions. Because heparin contains acidic mucopolysaccharides of different lengths, a layer of amino group can be added to the surface, which subsequently reacts with the polysaccharide chain end. The connecting procedure is straightforward and applicable to a wide variety of chemicals [194]. The biomedical sector is particularly interested in research into the design of heparin and heparin-like/mimicking polymer-functionalized biomedical materials, which is motivated by the potential for a wide range of biomedical applications such as HD membranes, cardiovascular stents, artificial organs, and other biomedical medical equipment. Cheng et al. [195] have published an overview of the most notable achievements in the field of surface heparinization. Surface coating is a facile and effective approach for heparinizing polymeric membranes, although its stability is not always adequate. The coated heparin stability can be increased by the electrostatic interaction between the negatively charged heparin and the positively charged polymer surface [196]. According to Huang et al. [197], heparin was covalently immobilized on the PSF membrane for specific adsorption of low-density lipoprotein (LDL). To produce accessible functional groups attached to heparin, the activation of PSF with successive chloromethyl ether and ethylenediamine treatments was required (Figure 12). The wettability of the PSF membrane was increased after heparin immobilization. Moreover, the heparinized PSF membrane considerably improved LDL adsorption in comparison to the pure PSF membrane. J. Li et al. [198] also reported heparin immobilization on the PSF surface via atmospheric pressure glow discharge (APGD). The disadvantage of using heparin is that it is exclusively obtained from animal tissues, posing a danger of virus infection and severe effects, such as thrombocytopenia for long-term therapies and hemorrhages in people receiving a high heparin dose. Heparin has also been demonstrated to block solely plasma-free thrombin but not clot-bound thrombin [199].

As a result, Heparin-mimic compounds have been developed in recent years due to improved structure, sulfation, and purity control. Recently, by one-step radiation-induced homogeneous polymerization of sodium styrene sulfonate, acrylic acid, and N-pyrrolidone, J. Wang et al. [200] modified PES using sulfonate and carboxylic functional groups to produce a heparin mimic structure. Flat-sheet and hollow fiber membranes were made, and the prepared membranes demonstrated excellent biocompatibility.

This research group in another work used in situ cross-linked copolymerizations of acrylic acid (AA) and 2-acrylanmido-2-methylpropanesulfonic acid to create heparin-mimicking PES membranes in both flat-sheet and hollow fiber geometries (AMPS) [44] (Figure 13a). H. Ji et al. [201] modified PES membranes using an in situ ring-opening reaction approach to create a heparin mimic layer on PES and produce a low-fouling, hemocompatible membrane. To begin, poly (glycidyl methacrylate) (PGMA) was introduced into PES membranes using a phase inversion technique and an in situ cross-linking polymerization. Next, by simply soaking the membranes in the PAA-AMPS solution, poly (acrylic acid-co-2-acrylanmido-2-methylpropanesulfonic acid) (PAA-AMPS) was covalently coated onto the PES membrane surfaces via an in situ ring-opening reaction between the PAA and PGMA. The protein attachment was lowered when compared to virgin PES membranes; while, the FRR and resistance to blood cell and bacterium adsorption grew dramatically (Figure 13b). The modified membrane demonstrated considerably better hemocompatibility, including self-anticoagulant characteristics, as a result of the heparin-mimicking modification.

All in all, Heparin as a linear glycosaminoglycan with high carboxylic and sulfonate groups is one of the most commonly used anticoagulants in HD therapy, and its anticoagulant ability is thought to be due to the functional groups. Heparin-like bulk modification has a stable modification impact as an old-fashioned modification technique. But using dangerous organic solvents, harsh reaction conditions, and time-consuming processes render this technique unsuitable. Furthermore, degradation, chain disruption, and side reactions are inevitable, all of which might have an impact on membrane mechanical characteristics [89,144].

#### 3.1.3. Plasma Technique

Almost any gas can be brought into a plasma state if it is given enough energy. Plasma is a combination of electrons, ions, and exciting species, including free radicals. Plasma modification is a versatile surface treatment that is widely employed to add chemically reactive functional groups to polymer surfaces in order to promote hydrophilicity and achieve low-fouling membrane surfaces [202]. Low-temperature plasma-induced grafting has been reported to increase the permeance and antifouling property of the PVDF and PES/PSF membranes by Z. Zhao et al. [203]. The activation of the polymer support by plasma (production of radicals) and the deposition of a new ZW layer on the membrane surface by polymerization are the two basic procedures in plasma-induced grafting of poly(zwitterion) chains on the membrane surface. The plasma gas composition and the applied processing variables can be changed to balance these effects. So, plasma variables such as power, pressure, treatment duration, sample distribution, polymerization settings (like monomer concentration), solvent selection, and grafting time can determine the amount of grafting and the length of zwitterionic chains. Therefore, the poly(zwitterionic) layer thickness can be reduced to the angstrom (Å) scale [204]. P. Salimi et al. [205] used SBMA monomers to modify the surface characteristics of the PES membrane to enhance the anti-fouling property utilizing the crona air plasma method. PES membranes were first made utilizing the phase inversion approach in this study. Following that, corona discharge initiation activated the PES membrane surface by peroxide groups, which was measured by DPPH (2, 2-diphenyl-1-picrylhydrazyl). The SBMA was covalently grafted onto the surface of the PES membrane, and PSBMA polymer was produced as a result (grafting- from method) (Figure 14a). For achieving an appropriate grafting layer to boost the membrane performance characteristics, four distinct applied powers and three different corona times were investigated. Changes in corona conditions resulted in a variety of grafting yields and thickness of the generated layer on the membrane surface, affecting permeation flux, fouling capacity, and other properties. A. Venault et al. [206] prepared a novel fouling resistance pseudo-zwitterionic PVDF membrane, using the surface grafting of 2-(methacryloyloxy) ethyl] trimethylammonium (TMA) and sulfopropyl methacrylate (SA) copolymer through glow dielectric barrier discharge (GDBD) plasma-induced surface copolymerization to improve the antifouling properties and the hemocompatibility of PVDF membrane. Through the initial molar content, it was possible to fairly control the membrane surface charge, and thus prepare either pseudo-zwitterionic membranes or membranes with positive or negative charge bias. The hemocompatibility of pseudo-zwitterionic membranes was outstanding (resistance to blood cells, proteins, and low activity of hemolysis).

S. H. Chen et al. [202] used plasma-induced surface zwitterionization to create PSBMA-grafted PP fibrous membranes with tunable grafting characteristics and hemocompatibility (Figure 14b). The correlation between the membrane blood compatibility with surface grafting structures, charge neutrality, hydrophilicity, and hydration capabilities was carefully evaluated. The results showed that applying a brush-like PSBMA layer to the PP membrane surface using atmospheric plasma is a hopeful method for preparing membranes with low cross-linking, good stability, and high balanced charge neutrality of PSBMA-grafted structures. N. Saxena et al. [207] showed that Argon–oxygen (Ar–O_2_) plasma treatment of PES membranes improved their hydrophilicity, resulting in decreased solute particle deposition and increased flow. Increased O_2_ contents and longer exposure times (60% O_2_ for 10 min) during plasma treatment had a major impact on the hydrophilic improvement of the surface.

The plasma technique can be used to alter polymer surfaces in several ways due to its multiple advantages, including (a) the outermost surface layer can be modified without affecting the bulk qualities; (b) plasma would influence the surface of any polymer; (c) based on the gas supplied, selecting the type of chemical modification for the polymer surface is simple; (d) using gas plasma, wet chemical issues such as residual solvent on the surface and swelling can be eliminated; and (e) uniform modification across the surface.

However, the requirement for vacuum technology, which has a substantial impact on the overall cost of the treatment and makes large-scale production difficult, is one of the main downsides of plasma treatment.

#### 3.1.4. Enzymatic Treatment

Enzymatic treatment is a kind of grafting technique for modifying the surface of membranes. An enzyme initiates surface hydrolysis and a chemical/electrochemical grafting reaction on the membrane [4]. By employing enzymes during the functionalization process, it will be possible to either directly introduce monomers/polymers to the membrane surface or create reactive free radicals, which are then reacted with the membrane via nonenzymatic reactions [208]. Using enzymes to modify membranes has various potential advantages. (1) enzymes have the ability to eliminate the need for reactive chemicals and toxic solvents in place to ensure safety; (2) enzymatic reactions are ecological friendliness since their selectivity can be used to remove the requirement for costly protection and deprotection processes, and (3) enzymes allow precise modification of macromolecular structure to improve polymer function.

N. Nady et al. proposed an environmentally friendly approach for grafting PES membranes using an enzyme-based method [209]. In this research, laccase was utilized to generate free radicals and graft phenolic acid monomers (such as 4-hydroxybenzoic acid) to the membrane. In comparison to more traditional procedures, the modification approach in this study was highly mild and ecologically friendly; it can be done at ambient temperature and requires just oxygen and water, with no hazardous chemicals. C. Amri et al. [210] developed alginate-based biopolymers with enhanced physical and chemical features after esterification with polyvinyl alcohol to be used as a biocompatible HD membrane. As a polymer modifier, PVA was selected for its high mechanical strength and biocompatibility, as well as its non-toxic nature. PVA hydroxyl groups should react with alginate carboxyl groups to generate an ester derivative. The esterification reaction is expected to improve the mechanical strength of the generated membrane. The hydrophobicity of the membrane increased after PVA modification, as evidenced by a decrease in the WCA. Low protein attachment and platelet adhesion showed that PVA-Alg membranes were more hemocompatible than native alginate membranes. Y. Dai et al. [211] used a PDA approach to graft argatroban (AG) and methoxy polyethylene glycol amine (mPEG-NH_2_) onto a non-thrombogenic PES dialyzer membrane. They successfully reduced heparin-induced thrombocytopenia by using PDA as an intermediate linker to graft AG to modify PES membrane. After immersing the PES substrates in an alkaline dopamine solution for 24 h, AG and mPEG-NH_2_ were covalently grafted onto the resulting membrane (Figure 15). Platelet adhesion and activation were suppressed, clotting times were prolonged, and thrombin production and complement activation were inhibited, demonstrating the modified membrane’s superior antithrombotic qualities.

However, there are also drawbacks to these green and sustainable technologies, including high costs and strict requirements (e.g., narrow working temperature window).

## 4. Conclusions and Perspectives

Our critical review has thoroughly covered the recent studies on membrane surface immobilization techniques and their impact on hemodialysis membrane hemocompatibility. Commercial membrane surface modification seeks to obtain the desired qualities and maintain its stability throughout a particular separation process. A modified membrane can also be used to overcome the shortcomings of the untreated membrane such as its low mechanical strength, poor hydrophilicity, fouling, and wetting. Therefore, we critically compared various methods of surface modification and their resulting hemocompatibility. In addition, we have summarized the influence of each modification technique on different ways of assessing hemocompatibility as summarized in Table 1, Table 2, Table 3 and Table 4 presented our overall analysis of the literature presented in this review paper. The majority of physical methods are straightforward, cost-effective, and environmentally friendly (no organic chemicals, toxic solvents, or residual substances), however, the improved surface qualities that these alterations provide are less stable than chemical methods. Chemical approaches, on the other hand, can lead to practically permanent membrane modification, but the modification reactions typically require organic solvents and chemicals. Only a few procedures, such as blending, coating, photochemical, and enzyme treatment, show similar advantages (facile, cheap, and eco-friendly). Blending, in particular, is a simple and effective strategy for improving both flat and hollow fiber membranes that may be used on large scale. The coating can boost membrane hydrophilicity and antifouling properties, but it usually diminishes pure water flux due to the inevitable deposition of the coating layer in membrane holes and microstructures. Furthermore, the coating stability throughout separation operations is a critical challenge, as the coating layer adheres to the membrane via non-covalent forces. Grafting and surface modification of polymer membranes using plasma, UV, ozone, and high-energy irradiation are efficient, facile, and controllable procedures for improving membrane chemical stability and performance during hemodialysis therapy. Nevertheless, their technical complication, environmentally friendly aspects, and sometimes high cost have limited their industrial-scale uses.

As can be seen in Figure 16, the coating is the most exploited technique to modify membrane surfaces in the last five years. The increased number of coating modifications can be explained by the fact that coating includes easier and lower-cost procedures to modify membrane surfaces. The dip-coating procedure, for instance, requires no additional equipment than standard laboratory equipment. It provides a wide variety of modifications by changing the solution in which the membrane is submerged. Several cross-links produced by grafting altered the hydrophobicity and hemocompatibility of the coated membranes. Due to the limited control during the experimental process, the coating technology has the potential to negatively impact the membrane thickness and porosity. It seems to be the most viable modification process for the industrial sectors since it does not involve high temperature, pressure, or energy for its design, resulting in the easiest fabrication process. Grafting ranks second among popular membrane modification procedures, possibly due to the enhanced stability of the resulting membrane, which is attributed to covalent bonding. The modification techniques trend in hemodialysis membrane technology has the same trend.

Because ZW polymers are an important category of the modified materials of HD membrane with improved fouling resistance to human serum protein deposition; in addition to enhanced biocompatibility properties, PES-SB and PSF-SB were used as models to study the impact of different modification methods on membrane hemocompatibility reported in the literature (Table 5).

Several factors can be used to evaluate the effectiveness of an approach, such as its ease of synthesis, chemical stability, the use of room-temperature procedures, and lower costs, the desired outcome and hemocompatibility are key parts of modification strategies in human health technology. Nevertheless, the most critical factor in defining membrane hemocompatibility is human serum protein adsorption, which elicits and catalyzes a complex cascade of biological responses in the HD process. Furthermore, the growing adherence to fibrinogen is one of the main reasons for platelet adhesion and even thrombosis. As a result, the key indicator for assessing the effect of different methods on hemocompatibility in this study was protein inhibition, which is typically observed as a result of ZW membrane immobilization (Figure 17). Based on Table 5 and Figure 17, the coating method shows the least amount of BFG absorption on the surface of SB modified membrane, which can be explained by its simple procedure and high grafting density of the ZWs on the membrane surface in comparison to the other methods. Even though grafting has a better antifouling ability than blending (Table 5, Figure 17), it cannot be applied to the entire as a definitive result because no comprehensive data or consistent results exist on the synthesis of a specific membrane using different methods under the same conditions.

WCA was used as the key indicator of hydrophilicity in Figure 18. As can be observed, the modified PES-SB by coating technique had the lowest WCA, which is in line with its higher fouling resistance.

Overall, extensive research has demonstrated that enhancing surface hydrophilicity, altering membrane roughness, and integrating charged groups on the surface would significantly decrease human serum protein adsorption. Despite the substantial understanding of the basic principle of surface modification techniques, however, it’s still a challenge to correlate the produced long-term antifouling qualities and hemocompatibility. Furthermore, the long-lasting stability of the modification part, homogeneity, shelf life, and its leaching from the membrane surface still issues.

The next generation of antifouling surfaces requires advanced modification technologies that allow for the construction of a precisely controlled three-dimensional structure of the grafted nanolayer. From the economical perspective, instead of replacing the entire membrane when fouling occurs, the grafted antifouling layer should be easily detached from the membrane surface (reversible fouling). As an outlook, the scaling-up of modification techniques and their implementation using actual process feed streams can be highlighted. Simpler methods, such as coating, will be likely used on commercial and industrial scales. Moreover, biomimetic membrane surfaces with novel antifouling capabilities are anticipated to be developed. Future modification approaches are likely to rely on green and aqueous solvents, resulting in environmentally friendly procedures.

## Figures and Tables

**Figure 1 membranes-12-01063-f001:**
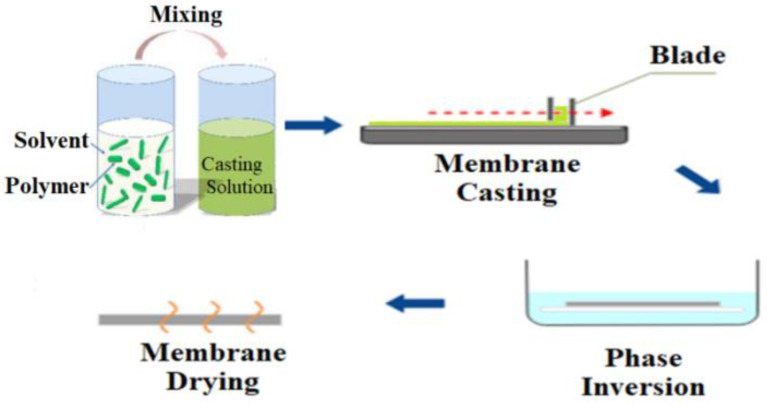
Modification of PSF membrane using the phase inversion method.

**Figure 2 membranes-12-01063-f002:**
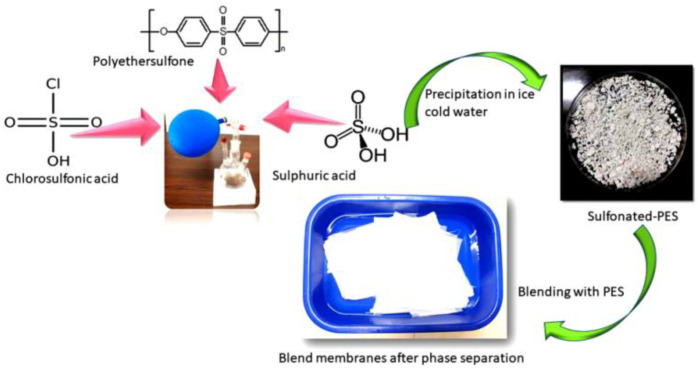
The synthetic procedure of SPES/PES blends membranes [81].

**Figure 3 membranes-12-01063-f003:**
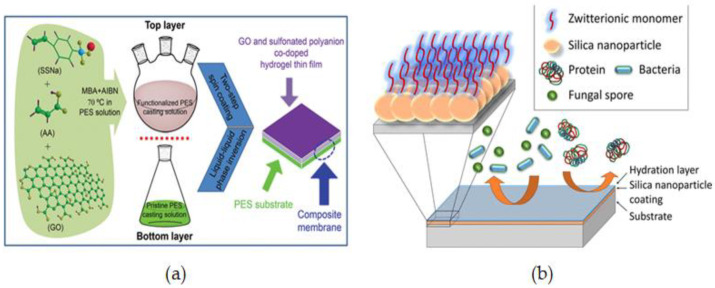
(**a**) Synthetic procedure of dual layer modified PES/GO-SPHF [101], (**b**) Surface modification of pre-deposited SiNPs functionalized with SB [102].

**Figure 4 membranes-12-01063-f004:**
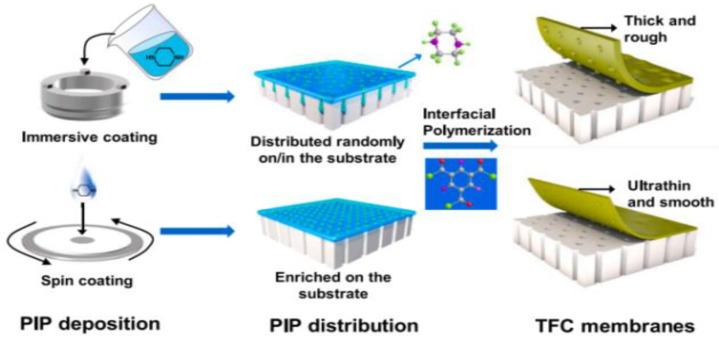
Schematically illustrated synthesizing PA membranes using IP and SCIP methods [108].

**Figure 5 membranes-12-01063-f005:**
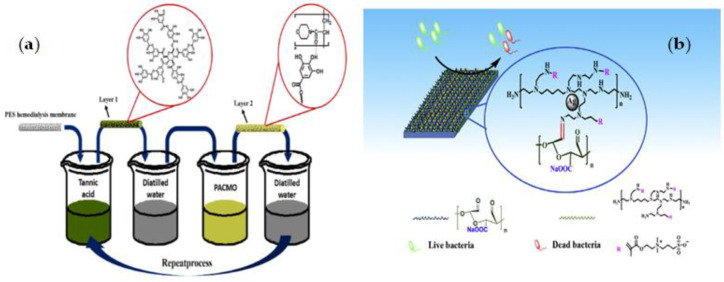
(**a**) The synthetic procedure ofPES/TA-PACMO via the LBL method [111]. (**b**) Preparation of PES/PEI-SBMA-Ag and PES/PEI-SBMA/OSA-n-Ag via Schiff-based LBL [112].

**Figure 6 membranes-12-01063-f006:**
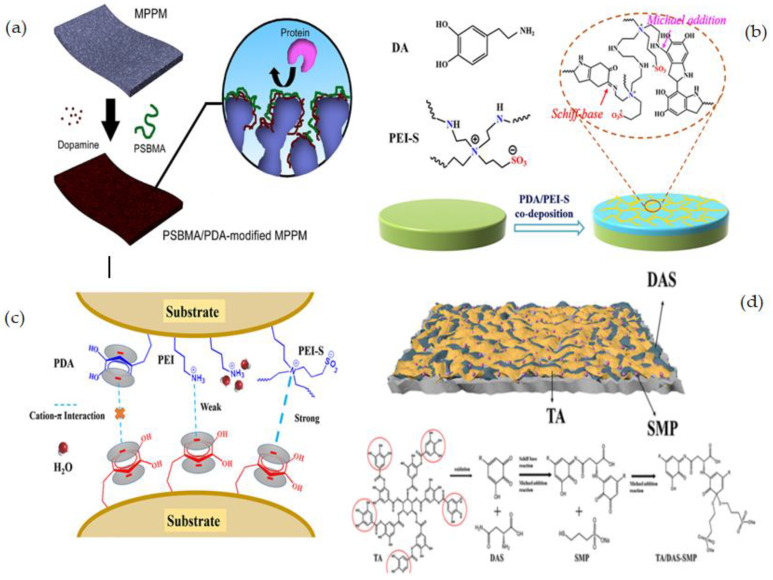
(**a**) Schematically illustration of co-deposition of PSBMA/PDA on MPPM membrane surface [120], (**b**) Synthetic procedure of PES/M-PDA/PEI-S [121], (**c**) Representative interaction between PDA, PEI, and PEI-S [121], (**d**) The reaction mechanism of TA/DAS-SMP [122].

**Figure 8 membranes-12-01063-f008:**
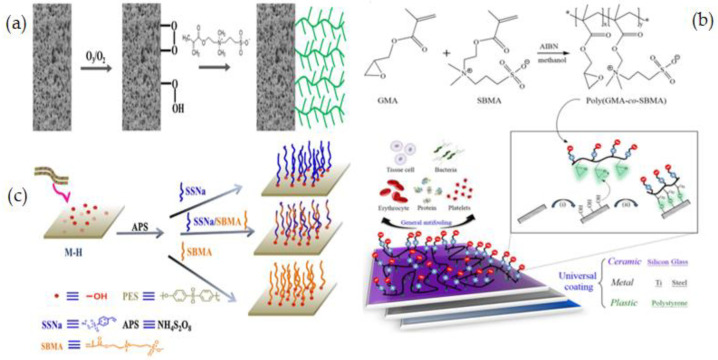
(**a**) Synthetic procedure of grafting ZW on PVDF membrane [137], (**b**) Synthesize and grafting to poly(GMA-co-SBMA) onto various surfaces via ultraviolet ozone pre-treatment and triethylamine (TEA) regulation of ring opening [140], (**c**) Schematic illustration of PES membrane functionalization [141].

**Figure 9 membranes-12-01063-f009:**
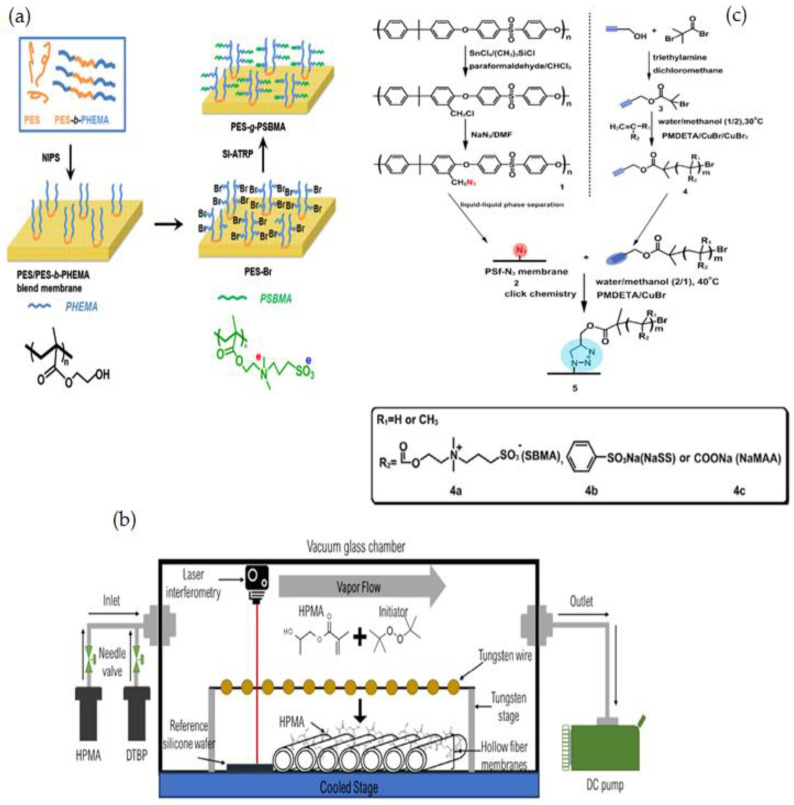
(**a**) Synthetic route of PES-g-PSBMA [152], (**b**) Schematically illustration of iCVD system [153], (**c**) Schematically preparation of PSF-N_3_, alkynyl-functionalized polymers, and modified PSF membrane by click chemistry [127].

**Figure 10 membranes-12-01063-f010:**
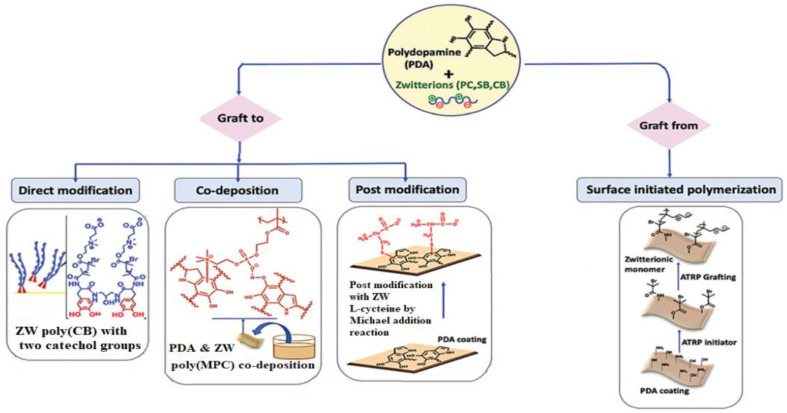
Different techniques of dopamine zwitterion conjugation [182].

**Figure 12 membranes-12-01063-f012:**
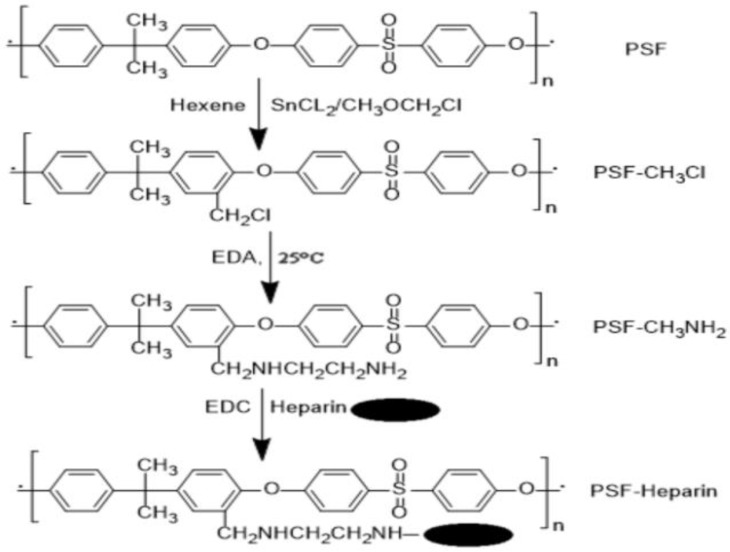
Immobilization of heparin on polysulfone membrane.

**Figure 13 membranes-12-01063-f013:**
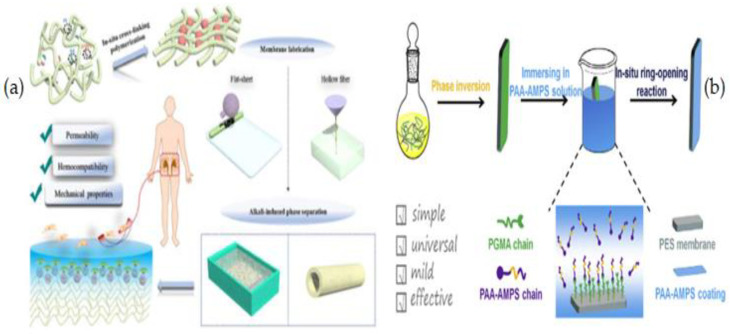
(**a**) Synthesis of heparin-mimicking semi-interpenetrating PES membranes via in situ cross-linked copolymerization [44], (**b**) Synthetic procedure of ES/GMA/PAA-AMPS membranes [201].

**Figure 14 membranes-12-01063-f014:**
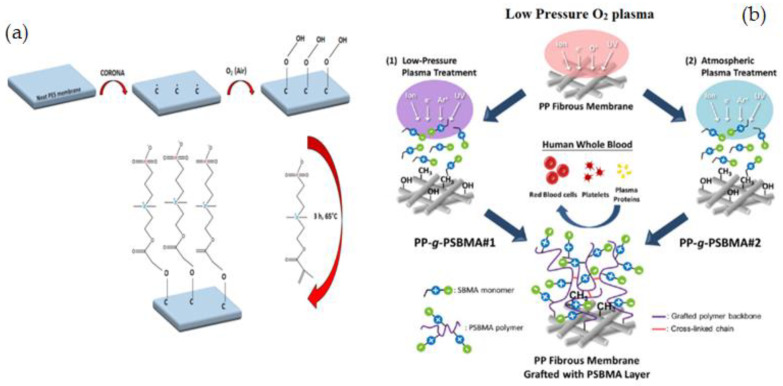
(**a**) ZW copolymerization process of PES membrane via corona air plasma [205], (**b**) Schematically synthesis of PSBMA-Grafted PP membranes using the low-pressure treatment and atmospheric plasma treatment [202].

**Figure 15 membranes-12-01063-f015:**
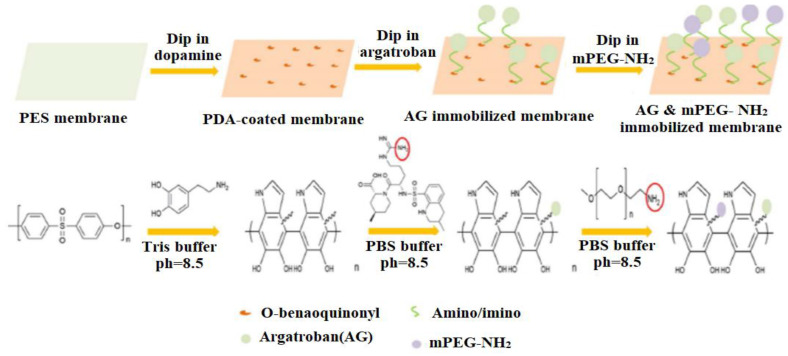
Schematically procedure of PES modification with PDA and AG/mPEG-NH_2_ [211].

**Figure 16 membranes-12-01063-f016:**
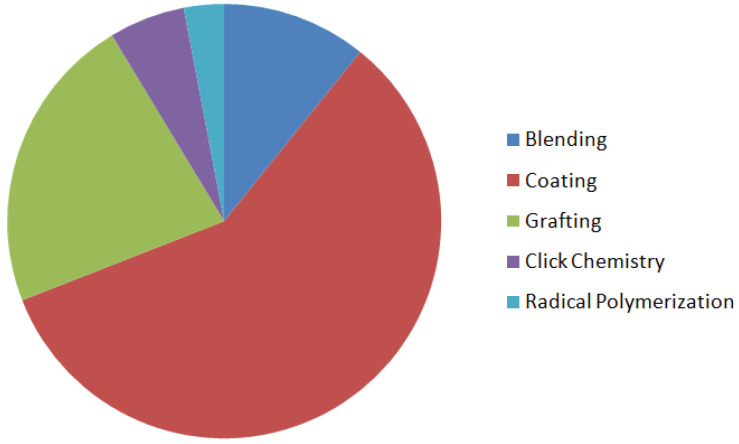
Statistical study of coating, blending, grafting, click chemistry, and radical polymerization for membrane surface modification in the last five years.

**Figure 17 membranes-12-01063-f017:**
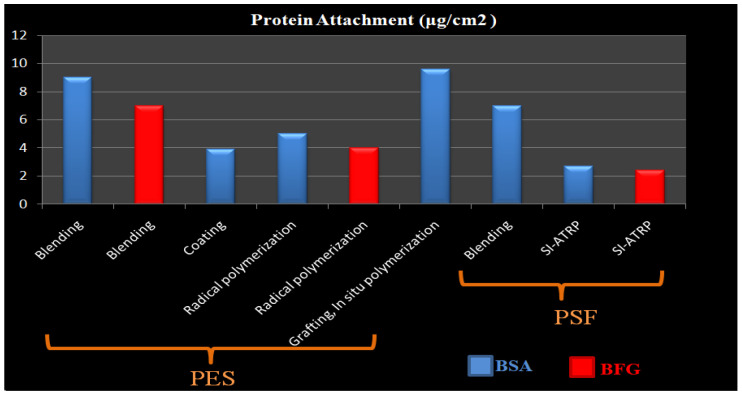
Comparison of protein attachment on modified SB zwitterionic-PES/PSF via different methods.

**Figure 18 membranes-12-01063-f018:**
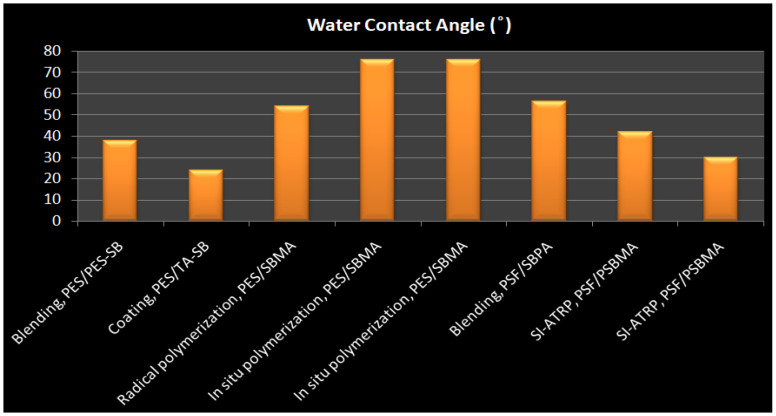
Comparison of WCA on modified SB Zwitterionic PES/PSF via different methods.

**Table 1 membranes-12-01063-t001:** Comparison of combining different additive blocks on membrane performance.

Name of Additive	Preparation Method of Modification Layer	wt% of Additive	Pure Water Flux (L m−2 h−1)	Contact Angle (θ°)	Antifouling Properties (%) FRR (BSA)	Protein Absorption	Platelet Adhesion	Clotting Time (PT, APTT)	Complement Activation	Ref.
**PLA-PHEMA**	RAFT	5, 10, 15, 20	236 for 15 wt% PLA-PHEMA content	71.4° to 60.5°	54.9 to 86.0%	28 to 2 μg/cm^2^	Suppressed platelet adhesion	Prolonged plasma recalcification times		[45]
**PEG-PVA**	Phase inversion method,	1. 1.5, 2, 2.5	16.4 to 42.484	37.3 to 46°	82 to 90%	low amount of protein	Suppressed platelet adhesion	Plasma recalcification times: 240 ± 5 s to300 ± 3 s	~5% ± 0.15 to ~6.5% ± 0.15 thrombus formation	[74]
**PVP-PES**	RAFT	1.11, 2.97, 4.50, 6.29, 8.12 and 9.47.	-	-	-	-	96% reduction in platelet adsorption.	APTT increased from 58 to 93 s.		[79]
**SPES**	Sulfonation	3		49.55 ± 1.52°		72 ± 5 ng/cm^2^	Decreased	178 ± 2 s	Minor amount of hemostatic fibrin	[81]
**MPC-PPGMA**	Non-solvent induced phase separation	3.1, 5.1, 6	80 L m^−2^ h^−1^		85–97%	BSA rejections are around 20–35%.	Total cell thickness decreased from about 0.43 µm^3^/μm^2^ on the pristine PVC membrane to 0.08 µm^3^/μm^2^			[85]
**PES-Aramid nanofiber**	Spin-coating method and phase inversion technique	0.5% CNT	Higher fluxes	62.5 ± 3.1°	92.3%	4.81 μg/cm^2^	3.1 × 10^7^ cells/cm^2^	Clearly improved		[84]
**PES and PSF Aramid nanofiber/Blending**	Spin-coating method and phase inversion technique	0.5% CNT	Higher fluxes	64.6 ± 2.0°	88.2%	5.03 μg/cm^2^	3.2 × 10^7^ cells/cm^2^	Clearly improved		[84]
**PVC-** **SBTFPU (20%)**		20%		similar level to pristine PVC (89.6◦)		2 μg/cm^2^ for BSA	0.84 × 10^6^ platelets/cm ^2^		Fibroblasts (NIH3T3 cell line) test, Higher than 90% cell viability	[86]

**Table 2 membranes-12-01063-t002:** Summary of selected surface-coating modification studies.

Membrane	Modification	Method	Main Results	Disadvantages	Ref.
**PP**	P(4VP-r-ODA)	SAMs	A versatile approach for developing bio-surfaces in vitro, Accurate control of the packing density and environment of an immobilized recognition center, or many centers, on a substrate surface.	The amphiphile concentration, pH, and ionic content all influence the self-assembly process.	[99]
**PVDF**	p(MAO-DMEA)	SAMs	Facile approach, no damage to bulk membrane properties, nanoscale control, Optimizing of coating variables (coating time, solution concentration), and modifier chemical composition (hydrophilic/hydrophobic ratio).	A two-step process, The thermal evaporation process was used as an additional processing stage that is time-consuming	[100]
**PES**	GO-SPHF	Spin-coating	Produce a thin, uniform coating, Controlling the film thickness, Novel fashion of dual-layered composite membranes with integrated advantages of GO and sulfonated polyanions, GO as a multifunctional nano building block linked to a variety of biomaterials, Maintaining membrane mechanical strength	Difficulty with large area samples. two-step spin coating is time-consuming, low efficiency of spin coating material (95–98% of the material is thrown away during the process),	[101]
**Gold**	SB-functionalized SiNPs	Spin-coating	Thickness can be adjusted easily by varying spin speed or viscosity of liquids. The ability of uniform thin film with low-cost production, Does not need the catalyst, Organize and control the chemistries at a materials’ interface, Relatively inexpensive technique, Quickly and easily deposit thin layers.	Depending on many different parameters make it a complex process, Difficulty with large area samples, Multilayer structure difficulty (more than 2 layers), Inability to control deposition accurately (homogeneity, roughness, etc.); Difficulty in making super-thin films (<10 nm).	[102]
**PP**	poly(MPC-random-BMA), poly(SBAArandom-BMA), poly(HEMA-random-BMA), PMPC-block-PBMA, PSBAA-block-PBMA, PHEMA-block-PBMA	Dip-coating	A simple and efficient method, minimal waste systems.	Difficult control of film thickness and surface roughness, Time consuming method	[103]
**PEI**		LBL	A versatile and simple technique for developing multilayer films with desired qualities, Formationofhighlystablemultilayersofzwitterionicpolymers	LBL is an appropriate method to produce multilayers of ZW polymers, under specific circumstances such as low pH or ionic strength.	[109]
**PSF, PDMS**	PEI/PAA-g-AZ/PEA-p(SBMA)	LBL	Easy & straightforward method, Undetectable nonspecific protein adsorption& completely inhibit platelet adhesion and L929 cells attachment	Limited by the substrate size, type, and shape, Possibility of different types of polymer surfaces	[110]
**PES**	TA-PACMO	LBL	Produce high blood compatibility surfaces, Reducing the danger of cell bursting and enhancing red blood cell retention during dialysis, Preventing oxidative stress and decreased complication risk	The risk of degradation during the performance as hydrogen bonding is not as strong as covalent bonding	[111]
**PES**	PEI-SBMA/OSA-n-Ag	LBL	The Schiff reaction is a strong approach in the biomedical area since it is simple, reversible, pH-sensitive, and biocompatible, as well as having a high stable modified membrane due to Schiff-based connections., Combination of ZW and Ag NPs also leads toantibacterial and antifouling surface	The possibility of association or dissociation of Schiff bases linkages due to different stimuli consisting of pH, vitamin B6 derivatives, amino acids, and enzymes	[112]
**PP**	SBMA/PDA	Co-deposition	A facile and efficient method, One-step process- Short reaction time process	Pore size reduction in a modified membrane, The co-deposition process is influenced by several parameters such as pH, temperature, solution concentration, and deposition time.	[120]
**PES**	M-PDA/PEI-S	Co-deposition	One-step mussel-inspired method, Simple, robust, and material-independent technique, Covalently anchors the PDA to the zwitterionic polymer to improve the coating stability, Producing excellent fouling resistance membrane surface.	Surface morphologies of the M-PDA/PEI and M-PDA/PEI-S noticeably altered compared to M-PDA, Pore size-reduction	[121]
**PES**	TA/DAS-SMP	Co-deposition	No changes to the membrane pores, Production of homogenous surfaces, Good hydrophilic properties, anti-pollution property, hemocompatibility and solute filtration capability, Excellent adhesive qualities, making it an excellent modification surface for materials such as PVDF, PP, PAN, etc.	The risk of degradation during the performance as hydrogen bonding is not as strong as covalent bonding	[122]

**Table 3 membranes-12-01063-t003:** Comparison of selected grafting methods.

Types of Surface Modification	Membrane	Monomer	Contact Angle (°) Before/After	Protein Adhesion	Platelet Adhesion	Advantages	Disadvantages	Ref.
“Grafting-to” method & coating	PDMS	(PGMA-co-SBMA)	117.93/79.51֩	2.89 μg/cm^2^ For FB (90% Fibrinogen adhesion reduction)	63.67% Platelet reduction	Non-destructive, simple, innovative method, Precise control of localized grafting, High chemical stability, Maintaining the basic properties of PDMS, such as its mechanical properties and transparency.	Not applicable for large-scale modification, Possibility of changing pore structure as a function of the grafting procedure	[128]
Bulk grafting method	PES	NHAc	76.6/48.1	-	-	An efficient technique with a stable modification impact due to chemical reactions, Providing membranes with long-term biocidal activity	Old fashioned and unsuccessful method due to the hazardous organic solvents, harsh environment, and time-consuming process	[130]
Thermal induced grafting	PES	CB & SB	10^◦^ reduced	66% reduction BSA	-	Facile, economic, and efficient method	Obtaining active sites usually necessitates chemical initiators as well as cleavage agents, Difficult process on a broad scale	[131]
Atmospheric plasma-induced surface copolymerization grafting	ePTFE	PSBMA	120°/22	Fibrinogen plasma protein was drastically reduced	Excellent resistance to platelet adhesion	Long-term stability because of covalent bonding, Flexible method in tailoring desired surface features with different monomers	Membrane structure and mechanical properties may be affected by plasma impact, and non-uniformity of modified membranes is possible	[133]
O_2_ plasma pre-treatment and UV-irradiatedgrafting	Polypropylene non-woven fabric (NWF)	MPDSAH	123/17	80% reduction for BSA	Excellent resistance to platelet adhesion	The rapid method that produces clean and uniform functional groups on the membrane surface, Accurate control of the packing density, Higher FRR, Excellent stability	Exactly control of different conditions including O2 plasma treatment time, UV irradiation, monomer concentration, etc.	[134]
Ozone induced grafting	PVDF	SBMA	-	Excellent BSA reduction	-	Efficient and simple procedure, Tuning the grafting density and chain length, No remarkablephysical and chemical changes on membrane properties	Excessive ozone leads toPVDF degradation and creates large pores	[137]
Ozone and UV-induced grafting to method	PS	Poly(GMA-co-SBMA	100/45	90% reduction of FB	89.3%, reduction of platelet adhesion, 28.6 * 10^4^ (cells per cm^2^	Simple, effective, and cost-effective grafting method, Producing stable chemisorption layer on various surfaces,	Control of several variables like the voltage, oxidation time, etc., UV can impact membrane structure and mechanical characteristics.	[140]
Radical graft polymerization	PES	P(SSNa-co-SBMA)	75/55	4.93 µg/cm^2^ BSA and 4.04 µg/cm^2^ Fibrinogen(BFG)	2 × 10^5^ cells/cm^2^	Convenient and versatile method, Adjusting of component ratio, Commercial and industrial potential for biomedical requirements.	Non-uniformity surface, the Possibility of changing pore structure as a function of the grafting procedure	[141]

**Table 4 membranes-12-01063-t004:** Qualitative comparison of different membrane modification methods.

Modification Method	Flux after Modification	Antifouling Property	Simplicity/Versatility	Chemical Stability	Functionalization	Eco-Friendly Process	Cost Effectiveness	Industrialization Potential
**Blending**								
**Coating**	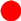			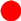				
**Grafting**						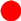		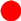
**Click chemistry**						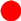		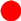
**Radical polymerization**						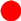		
**Plasma**			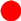			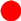	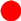	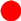
**Ozone**			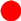			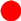		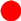
**Enzymatic treatment**								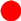

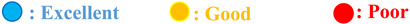

**Table 5 membranes-12-01063-t005:** Comparison of different modification methods on the hemocompatibility of PES-SB and PSF-SB membranes.

Membrane-ZW	Immobilization Method ZW	ZW Density(mg/cm^2^)	Water Contact Angle	Protein Adhesion	Platelet Adhesion	Pure Water Flux (L m^−^^2^ h^−^^1^)	Antifouling Properties (%) FRR *	Ref.
PES/PES-SB	Blending	-	37.8	9 μg/cm^2^ BSA (17% Reduction) & 7 μg/cm^2^ BFG	Significantly decreased	243.4	93.8%.	[212]
PES/PES-SBMA	Blending	-	59	90% BSA reduction	-	1157 ± 5.6	84%	[213]
PES/PSBMA	Blending	78% weight ratio	25.8 ± 4.6°	3.88 μm^3^/μm^2^ BSA	-	-	-	[214]
PES/DMMSA-BMA	Blending	-	48	>95% rejection of BSA	-		82.8%	[215]
PES/TA-SB (M-TA/PEI-S)	Dip-coating	-	24	3.9 μg/cm^2^ BFG	Very little			[216]
PES/PGMA-SB	Grafting, In situ cross-linking polymerization		43	0.60 lg/cm^2^ BSA and 0.37 l g/cm^2^ BFG	Suppressed platelet adhesion	43 mL/m^2^ h mmHg	100%	[217]
ES-b-PHEM/PSBMA	Grafting, SI-ATRP	High density	51	9 μg/cm^2^ BSA	Remarkably suppressed and almost no platelets adhered	39 L/m^2^ h	99%	[152]
PES/SBMA	Grafting, In situ cross-linking polymerizatio	High density	76	7 μg/cm^2^ BSA & 10 μg/cm^2^ BFG	Significantly decreased	705.21 mL/m^2^ h mmHg	99.11%	[218]
PES/SBMA	Radical graft polymerization	0.22	54	5 μg/cm^2^ BSA and 4 μg/cm BFG	2 × 10^5^ cells/cm^2^		-	[141]
PES/POEGMS-P(SBMA-co-AA)	LbL thiol-ene “click” chemistry	-	40	4.9 μg/cm^2^ BSA & 4.6 μg/cm^2^ BFG, 90% reduction	Nearly no adhered platelet	61	-	[219]
PSF/SB-PA	Blending	-	56.2	9.6 μg/cm^2^ BSA, 95% reduction	-	205 L/m^2^ h	85%	[220]
PSF/DEPAS	LBL & Click Chemistry	-	38	4 μg/cm^2^ BSA and 2μg/cm^2^ BFG,	Significantly decreased	-	-	[221]
PSF/PSBMA	Grafting, SI-ATRP	0.42 mg/cm^2^	42	2.7 μg/cm^2^ BSA& 2.4 μg/cm^2^ BFG	0.06 × 10^7^ cell/cm^2^	Decreased	Increased	[222]
PSF/PSBMA	Grafting	High density	30	98% Reduction	Significantly decreased	46.72	98.1%	[223]

* First cycle of FRR measurements was mentioned if there were multiple cycles.

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
