# Peer review of "Impact of Membrane Modification and Surface Immobilization Techniques on the Hemocompatibility of Hemodialysis Membranes: A Critical Review"

_membranes, 2022, doi:10.3390/membranes12111063_

Round 1

Reviewer 1 Report

This is a very extensive und interesting review on the impact of membrane modifications on hemocompatibility. The review covers the important aspect of protein adsorption, as this is a major determinant for reduction in hemocompatibility during hemodialysis treatments. Recently, dialysis membranes with increased PVP content / hydrophilicity on the blood-contacting surface were shown to have reduced protein adsorption and improved hemocompatibility in clinical studies. The present review is especially interesting for chemists and engineers working on membrane development. From a clinical perspective the title and abstract indicated more links to clinical applications. As this is not the case the authors should think to remove or rephrase lines 12-13 in the abstract, as the abstract should be a summary of the paper, which is not the case for this sentence. Is it correct that all mentioned membranes and modifications can be applied for (clinical) hemodialysis treatments? This is the expectation which is derived from the title. Many presented modifications are not possible to use for industrial production of commercially used dialyzers. The title and abstract should be clear that not only currently used membranes (most membranes applied for treatment are PS / PES – PVP membranes) are discussed but many possible modifications and basic research beyond clinical application. Maybe also Table 4 could include a column which modification has the potential to be applied for industrial use and the treatment of dialysis patients.

This is a comprehensive summary of the literature which is like a book chapter including many relevant information with regards to membrane modifications. Beyond that I have some minor comments, suggestions and questions:

Line 22-23: There is a formatting error

Figure on Page 2: This Figure does not have a legend. Moreover, the sense of the curly brackets is not clear. Also the formatting has some problems.

Line 34: Please shortly introduce protein adsorption to the membrane as membrane fouling before writing about antifouling properties. This wording may not be clear for all readers.

Line 38: What do you mean with efficiency? Performance?

Line 40/41: stability is mentioned twice

Line 52: References: There are other seminal papers on the impact of hydrophilic modification by PVP (e.g. Wang H, Yu T, Zhao C, Du Q. Improvement of hydrophilicity and blood compatibility on polyethersulfone membrane by adding polyvinylpyrrolidone. Fibers Polym. 2009;10:1–5; Hayama M, Yamamoto K, Kohori F, Sakai K. How polysulfone dialysis membranes containing polyvinylpyrrolidone achieve excellent biocompatibility? J Membrane Sci. 2004;234:41–9; Zhu L, Song H, Wang J, Xue L. Polysulfone hemodiafiltration membranes with enhanced anti-fouling and hemocompatibility modified by poly(vinyl pyrrolidone) via in situ crosslinked polymerization. Mater Sci Eng C Mater Biol Appl. 2017;74:159–66.)

General: The reference style is not in line with the journal guidelines and very difficult to check as neither the title nor the doi number is given.

Section 2.1: are some modifications presented here not also linked to thermal treatment?

Line 109/110: What does 18/3 and 18/6 mean? Is the same as 6/1 and 3/1 weight ratio?

Line 116/117: Interesting point; beside performance, could also be a reason for the better hemocompatibility profile seen with synthetic hemodialysis membranes sterilized with INLINE steam sterilization as compared to gamma sterilized dialyzers?

Line 149 / Figure legends: A detailed description of the figures would allow readers to faster understand the article.

Line 149: Text refers to citation 70, the figure legend to 71, which is correct?

Line 153: Reversible –> reversible

Line 172 and others: check wording for consistency; hemocompatibility vs biocompatibility

Line 207: The Figure between the three red arrows is blurred

Line 239/241: check use of abbreviations

Line 252 e.g.: check document for double blanks

Line 259: Formatting of table could be improved (e.g. Antifoulin-g) (also other tables, check font size etc)); moreover, to allow faster reading, the abbreviations could be additionally explained in the footnote of the table; sometimes it is not clear to which wt% the characterization is linked in the table (e.g. different wt% of the additive but only one result for platelet adhesion)

Line 341: Figure legend and figure would be better to have on the same page (also other figures)

Line 347: two commas

Line 376: superiority as compared to?

Line 551/552: check bracket

Line 600: check formatting of figure legend

Line 600: this figure has twice the link (a) and (b)

Line 786-806: check formatting

Line 815: abbreviation PS or PSF?

Line 840-851: check formatting

Line 886: Membranes à membranes

Line 894: (b) is missing in the figure

Line 905: check formatting of figure legend

Line 907-909: check formatting

Line 965: Figure: abbreviation PSu is underlined; should it be PS / PSF? / check formatting of legend

Line 987-995: Figure and legend should be on the same page

Line 1108-1114: check formatting

Line 1186: check formatting

Author Response

General Response:

The authors appreciate the reviewers’ recommendation and comments. Please note that all the scientific/technical pointed out by the reviewer, as comments and queries, within this manuscript have been thoroughly revised. Please, see the point-by-point responses to the reviewer’s comments below; and please, also refer to the applicable pages of the revised manuscript for confirmation. Sections within this manuscript are now presented with clarity; the correct reference formats have now been adopted, rearranged in citation and within the list. The changes were kept in the file with the track-change option for the editors and the reviewers kind attention and highlighted in a yellow color.

This is a very extensive und interesting review on the impact of membrane modifications on hemocompatibility. The review covers the important aspect of protein adsorption, as this is a major determinant for reduction in hemocompatibility during hemodialysis treatments. Recently, dialysis membranes with increased PVP content / hydrophilicity on the blood-contacting surface were shown to have reduced protein adsorption and improved hemocompatibility in clinical studies. The present review is especially interesting for chemists and engineers working on membrane development. From a clinical perspective the title and abstract indicated more links to clinical applications.

As this is not the case the authors should think to remove or rephrase lines 12-13 in the abstract, as the abstract should be a summary of the paper, which is not the case for this sentence. Is it correct that all mentioned membranes and modifications can be applied for (clinical) hemodialysis treatments?This is the expectation which is derived from the title. Many presented modifications are not possible to use for industrial production of commercially used dialyzers. The title and abstract should be clear that not only currently used membranes (most membranes applied for treatment are PS / PES – PVP membranes) are discussed but many possible modifications and basic research beyond clinical application.

Response: Thank you so much for the valuable comment. Some of the cited publications may have been reported in different application fields. So, in response to your comment, we revised the abstract (Page 1, Line 27). We didn't change the title because the primary focus and goal of this study are membrane modification for hemodialysis treatment.

Maybe also Table 4 could include a column which modification has the potential to be applied for industrial use and the treatment of dialysis patients.

Thank you for your useful comment. A column has been added (Page 11, Line 1227, Table 4).

This is a comprehensive summary of the literature which is like a book chapter including many relevant information with regards to membrane modifications. Beyond that I have some minor comments, suggestions and questions:

Line 22-23: There is a formatting error.

It was edited.

Figure on Page 2: This Figure does not have a legend. Moreover, the sense of the curly brackets is not clear. Also the formatting has some problems.

The graphical abstracthas been removed from the manuscript.

Line 34: Please shortly introduce protein adsorption to the membrane as membrane fouling before writing about antifouling properties. This wording may not be clear for all readers.

Thank you for your comment. It was added in line 36.

Line 38: What do you mean with efficiency? Performance?

Hemodialysis membrane efficiency and performance are defined by the degree of ultrafiltration to clearsolute and toxins from patients’ blood. The concept of high-efficiency dialysis is often defined by means of a high urea clearance rate.

Line 40/41: stability is mentioned twice

Thank you, it was removed.

Line 52: References: There are other seminal papers on the impact of hydrophilic modification by PVP (e.g. Wang H, Yu T, Zhao C, Du Q. Improvement of hydrophilicity and blood compatibility on polyethersulfone membrane by adding polyvinylpyrrolidone. Fibers Polym. 2009;10:1–5; Hayama M, Yamamoto K, Kohori F, Sakai K. How polysulfone dialysis membranes containing polyvinylpyrrolidone achieve excellent biocompatibility? J Membrane Sci. 2004;234:41–9; Zhu L, Song H, Wang J, Xue L. Polysulfone hemodiafiltration membranes with enhanced anti-fouling and hemocompatibility modified by poly(vinyl pyrrolidone) via in situ crosslinked polymerization. Mater Sci Eng C Mater Biol Appl. 2017;74:159–66.)

Many thanks for bringing these research works to my attention. Those references have been added in our manuscript. (Page 2, Line 52, Ref 24-32)

General: The reference style is not in line with the journal guidelines and very difficult to check as neither the title nor the doi number is given.

We appreciate the valuable comment. The reference style was revised based on the journal guidelines.

Section 2.1: are some modifications presented here not also linked to thermal treatment?

The authors appreciate your comment. It is correct. In section 2.1 (Reference 58), polishing and thermal treatment are coupled for surface modification. Surface modification generally involves several methods covering and complementing each other in order to produce a higher-quality surface.We can also see it in other techniques, such as coating and blending.

Line 109/110: What does 18/3 and 18/6 mean? Is the same as 6/1 and 3/1 weight ratio?

We appreciate the valuable comment. Yes, the weight ratios of 18/3 and 18/6 are the same as those of 6/1 and 3/1, however we wrote following literature style.

Line 116/117: Interesting point; beside performance, could also be a reason for the better hemocompatibility profile seen with synthetic hemodialysis membranes sterilized with INLINE steam sterilization as compared to gamma sterilized dialyzers?

We appreciate the reviewer’s interesting comment. This point was out of the scope of the study.

Line 149 / Figure legends: A detailed description of the figures would allow readers to faster understand the article.

Thank you for your comment. It was revised. (Page 4, Line 150)

Line 149: Text refers to citation 70, the figure legend to 71, which is correct?

The text refers to citation 70. Fig. 1 is created by the authors to demonstrate the phase inversion method.

Line 153: Reversible –> reversible

It was revised.

Line 172 and others: check wording for consistency; hemocompatibility vs biocompatibility

Thank you for mentioning this important issue. As the standard procedure of literature review, everything was reported based on the cited publication. Biocompatible is correct here based on the cited paper.

Line 207: The Figure between the three red arrows is blurred

We replaced it with a higher resolution figure.

Line 239/241: check use of abbreviations

It was done.

Line 252 e.g.: check document for double blanks

The manuscript has been checked for double blanks

Line 259: Formatting of table could be improved (e.g.Antifouling) (also other tables, check font size etc)); moreover, to allow faster reading, the abbreviations could be additionally explained in the footnote of the table; sometimes it is not clear to which wt% the characterization is linked in the table (e.g. different wt% of the additive but only one result for platelet adhesion)

Thank you for your useful comment. Formatting of the tables has been improved.

Line 341: Figure legend and figure would be better to have on the same page (also other figures)

Thanks for the great comment. It was revised as suggested.

Line 347: two commas

It was removed.

Line 376: superiority as compared to?

Thank you for your comment. This sentence was revised.(Page 9, Line 387-388)

Line 551/552: check bracket

It was completed.

Line 600: check formatting of figure legend

It was revised.

Line 600: this figure has twice the link (a) and (b)

It was revised

Line 786-806: check formatting

It was completed as requested.

Line 815: abbreviation PS or PSF?

It was corrected. (Line 877)

Line 840-851: check formatting

It was done.

Line 886: Membranes à membranes

It was corrected. (Page 25, Line 950)

Line 894: (b) is missing in the figure

It was corrected.

Line 905: check formatting of figure legend

It was done.

Line 907-909: check formatting

It was done.

Line 965: Figure: abbreviation PSu is underlined; should it be PS / PSF? / check formatting of legend

This figure was edited.

Line 987-995: Figure and legend should be on the same page

It was completed

Line 1108-1114: check formatting

It was completed, as requested

Line 1186: check formatting

It was done.

Reviewer 2 Report

Please make sure that many pictures in manuscript are belong to you (you creat by your self) so you dont need to write down the reference (cite the orifinal picture).

Organize size font and paragraph and picture and its explanation is shoud be in one page..

Author Response

Reviewer 2

Please make sure that many pictures in manuscript are belong to you (you create by yourself) so you dont need to write down the reference (cite the original picture).

Thank you for your comment. Some figures were redrawn as suggested.

Organize size font and paragraph and picture and its explanation is shoud be in one page.

All of the manuscript figures and tables were double-checked and formatted.

Reviewer 3 Report

The article is well organized and detailed. I recommend this paper for publication in its present form.

Author Response

Reviewer 3

The article is well organized and detailed. I recommend this paper for publication in its present form.

We appreciate the reviewer’s comments.
